# Genomics of cellular proliferation in periodic environmental fluctuations

Jérôme Salignon[†], Magali Richard[†] (iD), Etienne Fulcrand, Hélène Duplus-Bottin & Gaël Yvert[*] (iD)

## Abstract

**Living systems control cell growth dynamically by processing information from their environment. Although responses to a single environmental change have been intensively studied, little is known about how cells react to fluctuating conditions. Here, we address this question at the genomic scale by measuring the relative proliferation rate (fitness) of 3,568 yeast gene deletion mutants in out-of-equilibrium conditions: periodic oscillations between two environmental conditions. In periodic salt stress, fitness and its genetic variance largely depended on the oscillating period. Surprisingly, dozens of mutants displayed pronounced hyperproliferation under short stress periods, revealing unexpected controllers of growth under fast dynamics. We validated the implication of the high-affinity cAMP phosphodiesterase and of a regulator of protein translocation to mitochondria in this group. Periodic oscillations of extracellular methionine, a factor unrelated to salinity, also altered fitness but to a lesser extent and for different genes. The results illustrate how natural selection acts on mutations in a dynamic environment, highlighting unsuspected genetic vulnerabilities to periodic stress in molecular processes that are conserved across all eukaryotes.**

**Keywords** fitness; fluctuating environment; selection; stress; yeast
**Subject Categories** Genome-Scale & Integrative Biology; Quantitative Biology & Dynamical Systems; Signal Transduction
**Mol Syst Biol. (2018) 14: e7823**

## Introduction

Cells are dynamic systems that modify themselves in response to variation of their environment. Interactions between internal dynamics of intracellular regulations and external dynamics of the environment can determine whether a cell divides, differentiates, cooperates with other cells or dies. For some systems, usually from model organisms, the molecules involved in signal transduction and cellular adaptation are largely known. How they act in motion, however, is unclear, and it is difficult to predict which

ones may be essential upon certain frequencies of environmental fluctuations. In addition, since most molecular screens were conducted in steady stress conditions or after a single stress occurrence, molecules that are key to the response dynamics may have been missed.

The control of cellular proliferation is essential to life and is therefore the focus of intense research, but the interplay between proliferative control and environmental dynamics remains poorly characterized. In addition, proliferation drives evolutionary selection, and the properties of natural selection in fluctuating environments are largely unknown. Although experimental data exist (Stomp *et al*, 2008; Bleuven & Landry, 2016), they are scarce and the selection of mutations in dynamic conditions has mostly been studied under theoretical frameworks (Kussell & Leibler, 2005; Cvijović *et al*, 2015; Sæther & Engen, 2015; Svardal *et al*, 2015). Repeated stimulations of a cellular response may have consequences on growth that largely differ from the consequences of a single stimulus. First, a small growth delay following a stimulus may become highly significant when cumulated over multiple stimuli. Second, growth rate at a given time may depend on past environmental conditions that cells "remember", and this memory can sometimes be transmitted to daughter cells (Hilker *et al*, 2016). These two features are well illustrated by the study of Razinkov *et al*, who manipulated the stability of GAL1 mRNA transcripts that participated to short-term "memory" of galactose exposures: this resulted in a growth delay that was negligible after one galactose-to-glucose change but significant over multiple changes (Razinkov *et al*, 2013). Other memorization effects were observed in bacteria during repeated lactose to glucose transitions, this time due to both short-term memory conferred by persistent gene expression and long-term memory conferred by protein stability (Lambert & Kussell, 2014).

The yeast response to high concentrations of salt is one of the best-studied mechanisms of cellular adaptation. When extracellular salinity increases abruptly, cell size immediately reduces and yeast triggers a large process of adaptation. The translation programme (Uesono & Toh-e, 2002; Warringer *et al*, 2010) and turnover of mRNAs (Miller *et al*, 2011) are re-defined, calcium accumulates in the cytosol and activates the calcineurin pathway (Ariño *et al*, 2010), osmolarity sensors activate the high-osmolarity glycerol MAPK pathway (Hohmann, 2009; Ariño *et al*, 2010), glycerol

---

Laboratory of Biology and Modeling of the Cell, Ecole Normale Supérieure de Lyon, CNRS, Université Claude Bernard de Lyon, Université de Lyon, Lyon, France
*Corresponding author. Tel: +33 4 72 72 80 00; E-mail: gael.yvert@ens-lyon.fr
†These authors contributed equally to this work

accumulates intracellularly as a harmless compensatory solute (Hohmann, 2009), and membrane transporters extrude excess ions (Ariño *et al*, 2010). Via this widespread adaptation, hundreds of genes are known to participate to growth control after a transition to high salt. What happens in the case of multiple osmolarity changes is less clear, but can be investigated by periodic stimulations of the adaptive response. For example, periodic transitions between 0 and 0.4 M NaCl showed that MAPK activation was efficient and transient after each stress except in the range of ~8-min periods, where sustained activation of the response severely hampered cell growth (Mitchell *et al*, 2015). How genes involved in salt tolerance contribute to cell growth in specific dynamic regimes is unknown.

Yeast cells also respond to extracellular methionine concentrations by modulating its import and biosynthesis. This response relies on molecular pathways that are largely unrelated to salt stress. It has also been well described (Thomas & Surdin-Kerjan, 1997), but not in the context of dynamic environmental fluctuations.

We sought to systematically search for genes involved in the dynamics of a cellular response. Identifying such genes can be done by applying specific stimulations to mutant cells periodically and testing whether the effect of the mutation on proliferation is averaged over time. In other words, does fitness (proliferation rate relative to wild type) of a mutant under periodic stress match the time average of its fitness in each alternating condition? This problem of temporal heterogeneity is equivalent to the homogenization problem commonly encountered in physics for spatial heterogeneity, where microscopic heterogeneities in materials modify macroscopic properties such as stiffness or conductivity (Hassani & Hinton, 1998). A homogeneous fitness (averaged over time) implies that (i) the effect of a mutation on the response occurs rapidly as compared to the frequency of environmental changes, (ii) it does not affect the response lag phase and (iii) the mutated gene is not involved in relevant memory mechanisms. In contrast, fitness inhomogeneity (deviation from time-average expectation) is indicative of a role of the gene in the response dynamics.

In this study, we present two genomic screens that address this homogenization problem in the context of yeast cells responding to periodic salt stress or periodic methionine availability. The results reveal how selection of mutations can depend on environmental oscillations and identify molecular processes that unexpectedly become major controllers of proliferation at short periods of repeated stress.

# Results

## Genomic profiling of proliferation rates in steady and periodic salt stress

We measured experimentally the contribution of thousands of yeast genes on proliferation in two steady conditions of different salinity and in an environment that periodically oscillated between the two conditions. We used a collection of yeast mutants where ~5,000 non-essential genes have been individually deleted (Giaever *et al*, 2002). Since every mutant is barcoded by a synthetic DNA

tag inserted in the genome, the relative abundance of each mutant in pooled cultures can be estimated by parallel sequencing of the barcodes (BAR-Seq) (Smith *et al*, 2009; Robinson *et al*, 2014). We set up an automated robotic platform to culture the pooled library by serial dilutions. Every 3 h (average cell division time), populations of cells were transferred to a standard synthetic medium containing (S) or not (N) 0.2 M NaCl. The culturing programme was such that populations were either maintained in N, maintained in S or exposed to alternating N and S conditions at periods of 6, 12, 18, 24 or 42 h (Fig 1A). Every regime was run in quadruplicates to account for biological and technical variability. The duration of the experiment was 3 days, and populations were sampled at times 0, 24, 48 and 72 h for sequencing. After data normalization and filtering, we examined how relative proliferation rates compared between the periodic and the two steady environments.

## Protective genes have diverse contributions to proliferation under periodic stress

We observed that genes involved in salt tolerance during steady conditions differed in the way they controlled growth under periodic regimes. Differences were visible both among genes inhibiting growth and among genes promoting growth in high salt. For example, NBP2 is a negative regulator of the HOG pathway (Mapes & Ota, 2004) and MOT3 is a transcriptional regulator having diverse functions during osmotic stress (Montañés *et al*, 2011, 3; Martínez-Montañés *et al*, 2013). As shown in Fig 1B, deletion of either of these genes improved tolerance to steady 0.2 M NaCl (condition S). In the 6-h periodic regime, the relative growth of *mot3Δ/Δ* cells was similar to the steady condition N, as if transient exposures to the beneficial S condition had no positive effect. In contrast, the benefit of transient exposures was clearly visible for *nbp2Δ/Δ* cells. Differences were also apparent among protective genes. The Rim101 pathway has mostly been studied for its role during alkaline stress (Ariño *et al*, 2010), but it is also required for proper accumulation of the Ena1p transporter and efficient Na$^+$ extrusion upon salt stress (Marqués *et al*, 2015). Eight genes of the pathway were covered by our experiment. Not surprisingly, for all positive regulators of the pathway, gene deletion decreased proliferation in S and increased proliferation in N (Fig 1B and Appendix Fig S1). This is consistent with the need of a functional pathway in S and the cost of maintaining it in N where it is not required. However, the response to 6-h periodic stimulations was different between mutants (Appendix Fig S1). Although RIM21, DFG16 and RIM9 all code for units of the transmembrane sensing complex (Obara *et al*, 2012), proliferation was high for *rim21Δ/Δ* and *dfg16Δ/Δ* cells but not for *rim9Δ/Δ* cells. Similarly, Rim8 and Rim20 both mediate the activation of the Rim101p transcriptional repressor (Xu & Mitchell, 2001; Herrador *et al*, 2010); but *rim8Δ/Δ* and *rim101Δ/Δ* deletions increased proliferation under periodic stress, whereas *rim20Δ/Δ* did not. This pathway was not the only example displaying such differences. Cells lacking either the HST1- or the HST3 NAD(+)-dependent histone deacetylase (Brachmann *et al*, 1995) grew poorly in S, but *hst1Δ/Δ* cells tolerated 6-h periodic stress better than *hst3Δ/Δ* cells (Fig 1B).

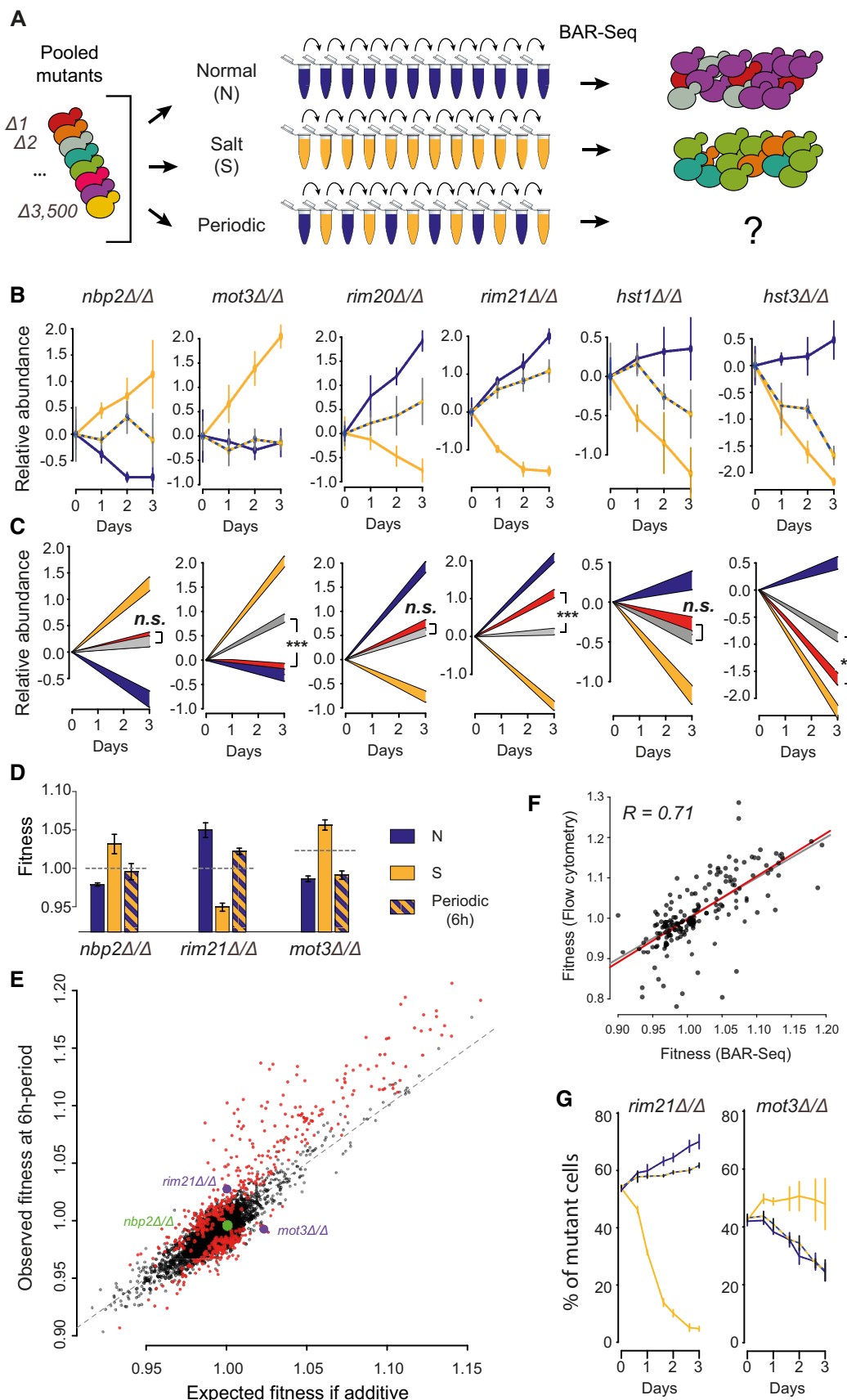

**Figure 1.**

**Figure 1.  Genomic profiling of fitness in periodic salt stress.**

A   Experimental design. Populations of yeast deletion strains are cultured in media N (no salt), S (salt) and in conditions alternating between N and S at various periods. Allele frequencies are determined by BAR-Seq and used to compute fitness (proliferation rate relative to wild type) of each mutant.

B   Time course of mutant abundance in the population, shown for six mutants. Relative abundance corresponds to the median of $log_2(y/y_0)$ values $\pm$ SD ($n = 4$ replicate cultures, except for condition N at day 3: $n = 3$), where $y$ is the normalized number of reads, and $y_0$ is $y$ at day 0. Conditions: N (blue), S (yellow), 6-h periodic oscillations (NS6, hatching).

C   Generalized linear models (*predicted value* $\pm$ SE) fitted to the data shown in (B), coloured by condition: N (blue); S (yellow); NS6 predicted by the null model (grey) or predicted by the complete model including inhomogeneity (red). ***$P < 10^{-8}$. *n.s.*, non-significant, based on the GLM (see Materials and Methods).

D   Fitness values ($\omega$) computed from the data of two mutants shown in (B). Bars, mean $\pm$ SEM, $n = 3$ (N) or 4 (S, NS6) replicate cultures, coloured according to culture condition. Grey dashed line: expected fitness in case of additivity (geometric mean of fitness in N and S weighted by the time spent in each medium).

E   Scatterplot of all mutants showing their observed fitness under 6-h periodic oscillations (*y*-axis, NS6 regime) and their expected fitness in case of additivity (*x*-axis, weighted geometric mean of fitness in N and S). Deviation from the diagonal reflects inhomogeneity. Red dots: 456 mutants with significant inhomogeneity (FDR = 0.0001, see Materials and Methods).

F   Correlation between fitness estimates ($\omega$). Each dot corresponds to the median fitness of one mutant in one condition (N, S or NS6), measured from pooled cultures (*x*-axis) or from individual assays (one mutant co-cultured with WT cells, *y*-axis). Whole data: 52 mutants. *R*, Pearson coefficient; grey line, *y = x*; red line, linear regression.

G   Validation of inhomogeneity by cell counting. One graph shows the time course of mutant abundance when it was individually co-cultured with GFP-tagged wild-type cells, measured by flow cytometry. Median values $\pm$ SD ($n = 4$ replicate cultures). Conditions: N (blue), S (yellow), 6-h periodic oscillations (NS6, hatching).

Thus, gene deletion mutants of the same pathway or with similar fitness alterations in steady conditions can largely differ in their response to dynamic conditions.

**Widespread deviation from time-average fitness in periodic salt stress**

We then systematically asked for each of the 3,568 gene deletion mutants, whether its fitness in periodic salt stress matched the time average of its fitness in conditions N and S. We tested both the statistical significance and quantified the deviation from the time-average expectation. For statistical inference, we exploited the full BAR-Seq count data, including all replicated populations, by fitting to the data a generalized linear model that included a non-additive term associated with the oscillations (see Materials and Methods). The models obtained for the six genes discussed above are shown in Fig 1C. Overall, at the 6-h period, we estimated that deviation from time-average fitness was significant for as many as ~2,000 genes, because it was significant for 2,497 genes at a false discovery rate (FDR) of 0.2 (Appendix Table S1). At a stringent FDR of 0.0001, we listed 456 gene deletions for which fitness inhomogeneity was highly significant.

For quantification, we computed fitness values as in Qian *et al* (2012) (Fig 1D) and plotted the observed fitness of all genes in the 6-h periodic environment as a function of their expected time-average fitness (Fig 1E). As for *nbp2Δ/Δ*, observed and expected values were often in good agreement. Highlighting the 456 significant genes revealed a surprising trend: for the majority of gene deletions expected to increase proliferation in the periodic regime (expected fitness > 1), observed fitness was unexpectedly high. Gene annotations corresponding to higher-than-expected fitness were enriched for transcriptional regulators and for members of the cAMP/PKA pathway (Appendix Table S2), which is consistent with cellular responses to environmental dynamics.

Although BAR-Seq can estimate thousands of fitness values in parallel, it has two important limitations: estimation by sequencing is indirect and the individual fitness of a mutant is not distinguished from possible interactions with other mutants of the pool. We therefore sought to validate a subset of our observations by applying individual competition assays. Each mutant was co-cultured with a GFP-tagged wild-type strain, in N or S conditions or under the 6-h periodic regime, and the relative number of cells was counted by flow cytometry (Qian *et al*, 2012; Duveau *et al*, 2014). Correlation between fitness estimates from BAR-Seq and individual assays was similar to previous reports (Qian *et al*, 2012; Venkataram *et al*, 2016; Fig 1F, Appendix Fig S2), and the assays unambiguously validated the fitness inhomogeneity of several mutants including *rim21Δ/Δ* and *mot3Δ/Δ* (Fig 1G).

**Impact of salinity dynamics on mutant proliferation**

If fitness inhomogeneity (deviation from time average) is attributable to environmental dynamics, then it should be less pronounced at large periods of oscillations. Our experiment included four conditions with periods larger than 6 h. For each period, we computed for each mutant the ratio between its observed fitness in periodic stress and the time-average expectation from its fitness in the two steady conditions N and S. Fitness is inhomogeneous when this ratio deviates from 1. Plotting the distribution of this ratio at different periods of oscillation showed that, as expected, inhomogeneity was less and less pronounced as the period increased (Fig 2A). We examined more closely three mutants displaying the highest inhomogeneity at the 6-h period. Plotting their relative abundance in the different populations over the time of the experiment clearly showed that fitness of these mutants was unexpectedly extreme at short periods but less so at larger periods (Fig 2B).

The result showing that some mutants but not all were extremely fit to short-period oscillations suggested that the extent of differences in fitness between mutants may change with the environmental period. To see whether this was the case, we estimated the genetic variance in fitness of each pooled population of mutants. Distinguishing the genetic variance from the non-genetic variance was possible because of the presence of replicates in our experimental design (see Materials and Methods). Fitness variation between strains was more pronounced when populations were grown in S than in N, which agrees with the known effect of stress on fitness differences (Martin & Lenormand, 2006). Remarkably, differences were even larger in fast-oscillating regimes, but not slow-oscillating ones (Fig 2C). This shows that environmental dynamics can exert additional selective pressures at the level of the whole population (see Discussion).

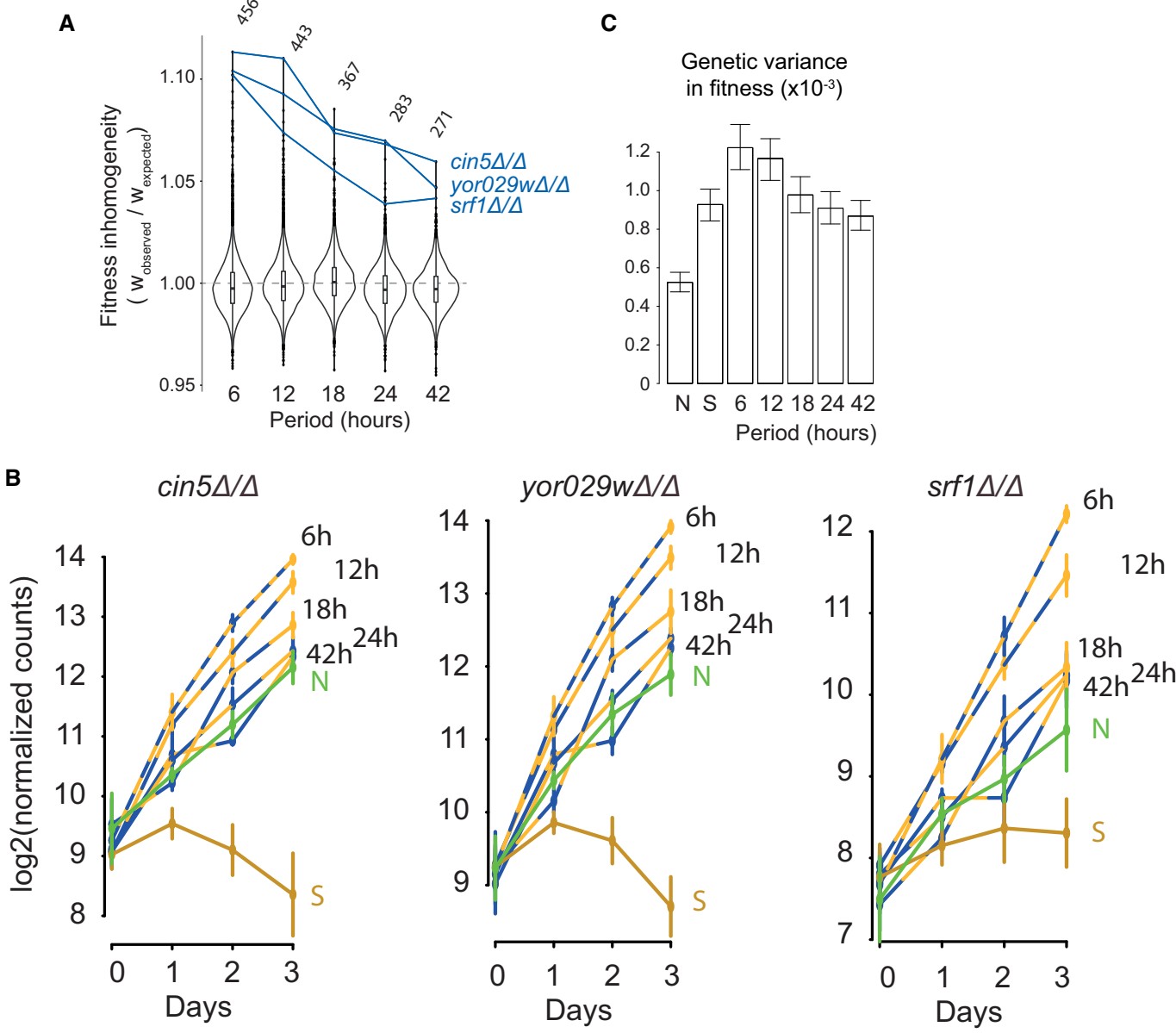

**Figure 2.   Proliferative advantage depends on environmental dynamics.**

A  Violin plots showing the distribution of fitness inhomogeneity of 3,568 gene deletions at the indicated periods of salinity oscillations. Traces and labels, mutants with extreme inhomogeneity at 6-h period. Top, number of gene deletions with significant inhomogeneity at FDR = 0.0001.

B  Time course of the abundance of mutants *cin5Δ/Δ*, *srf1Δ/Δ* and *yor029wΔ/Δ* in the pool of all mutants, under different alternating regimes, quantified by BAR-Seq. Median values ± SD (*n* = 4 replicate cultures, except for the N condition at day 3: *n* = 3).

C  The genetic variance in fitness of the pooled population of mutants was computed for each condition. Bars: 95% CI bootstrap intervals.

## Fitness in periodic salt stress vs. steady conditions

We asked whether fitness inhomogeneity in the 6-h periodic stress was related to fitness values in steady conditions, and for many gene deletions, it was. Inhomogeneity was associated with high fitness in both N and S conditions (Fig 3A–D, red dots). Interestingly, a particular set of gene deletions displayed very high fitness inhomogeneity together with distinct fitness in steady conditions: advantageous in N but not in S (Fig 3A–D, blue dots). Annotations of these genes were enriched for osmosensing and response

(Appendix Table S2), although several gene deletions of this functional category did not display this behaviour. Thus, cells defective for specific components of the osmostress response can outproliferate other cells in this periodic environment, likely because they do not trigger a costly and unnecessary adaptive process.

Some gene deletions improved growth in one steady condition ($w > 1$) and penalized it in the other ($w < 1$). This phenomenon is a special case of gene × environment interaction called antagonistic pleiotropy (AP) (Qian *et al*, 2012). It is difficult to anticipate whether such mutations will have a positive or negative impact on

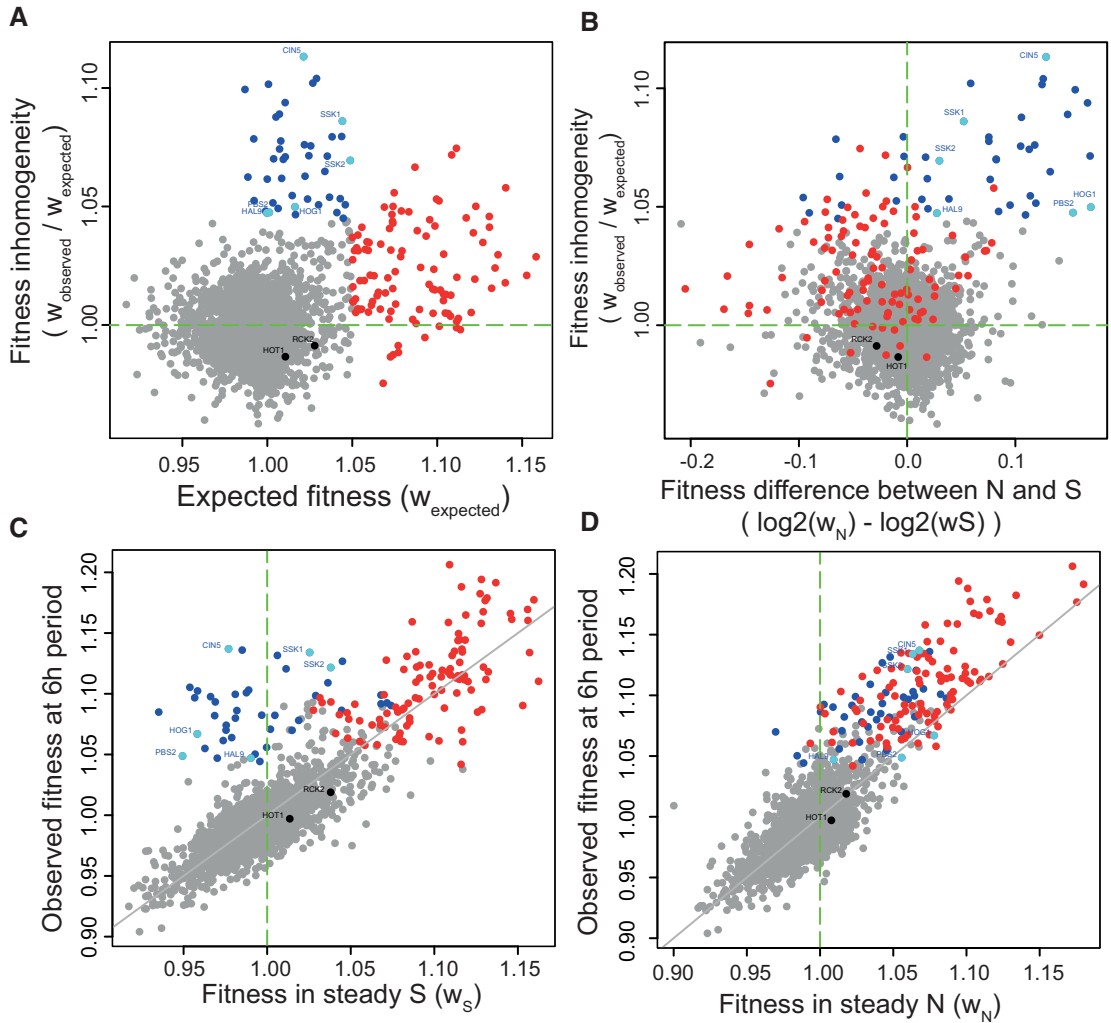

**Figure 3.  Fitness inhomogeneity with respect to fitness in steady conditions.**

A    Fitness inhomogeneity in the 6-h periodic regime is poorly correlated to expected fitness. Red, genes with high expected fitness (> 1.05). Blue, genes with high inhomogeneity (> 1.045) but moderate expected fitness (< 1.05), with those annotated in relation to osmosensing and response shown in light blue and by name. Black dots, examples of genes involved in osmosensing/response, but which do not display inhomogeneity.

B    Fitness inhomogeneity vs. difference in fitness between the two steady environments. Colours, same as panel (A).

C, D  Observed fitness in the 6-h periodic regime as a function of fitness in S (panel C) or fitness in N (panel D). Grey line: identity. Colours, same as in panel (A).

growth in a periodic regime that alternates between favourable and unfavourable conditions, especially since fitness is not necessarily homogenized over time. Using a specific test (see Materials and Methods), we found 48 gene deletions with statistically significant AP between the N and S conditions (FDR = 0.01, Appendix Table S3 and Fig S3). Interestingly, three of these genes coded for subunits of the chromatin-modifying Set1/COMPASS complex (Appendix Table S2 and Fig S4). We investigated whether the direction of effect of these 48 deletions depended on the period of oscillations (Fig 4A–C). The effect was positive at all periods for 33 AP deletions and negative at all periods for 6 AP deletions. For two mutations (*vhr1Δ/Δ* and *rim21Δ/Δ*), the direction of selection changed with the oscillating period. To visualize the periodicity dependence of all AP deletions, we clustered them according to their fitness inhomogeneities (Fig 4B and C). This highlighted five different behaviours:

oscillations could strongly favour proliferation of a mutant at all periods (e.g. *cin5Δ/Δ*) or mainly when they were fast (e.g. *oca1Δ/Δ*), they could mildly increase (e.g. *rim101Δ/Δ*) or decrease it (e.g. *csf1Δ/Δ*) or they could both increase and decrease it depending on their period (*vhr1Δ/Δ*). Thus, fitness during alternating selection was generally asymmetric in favour of positive selection, and its dependency to the alternating period differed between genes.

## Salinity oscillations heighten the proliferation of some mutant cells

We made the surprising observation that fitness during salt oscillations could exceed or fall below the fitness observed in both steady conditions (Fig 2B), a behaviour called "*transgressivity*" hereafter.

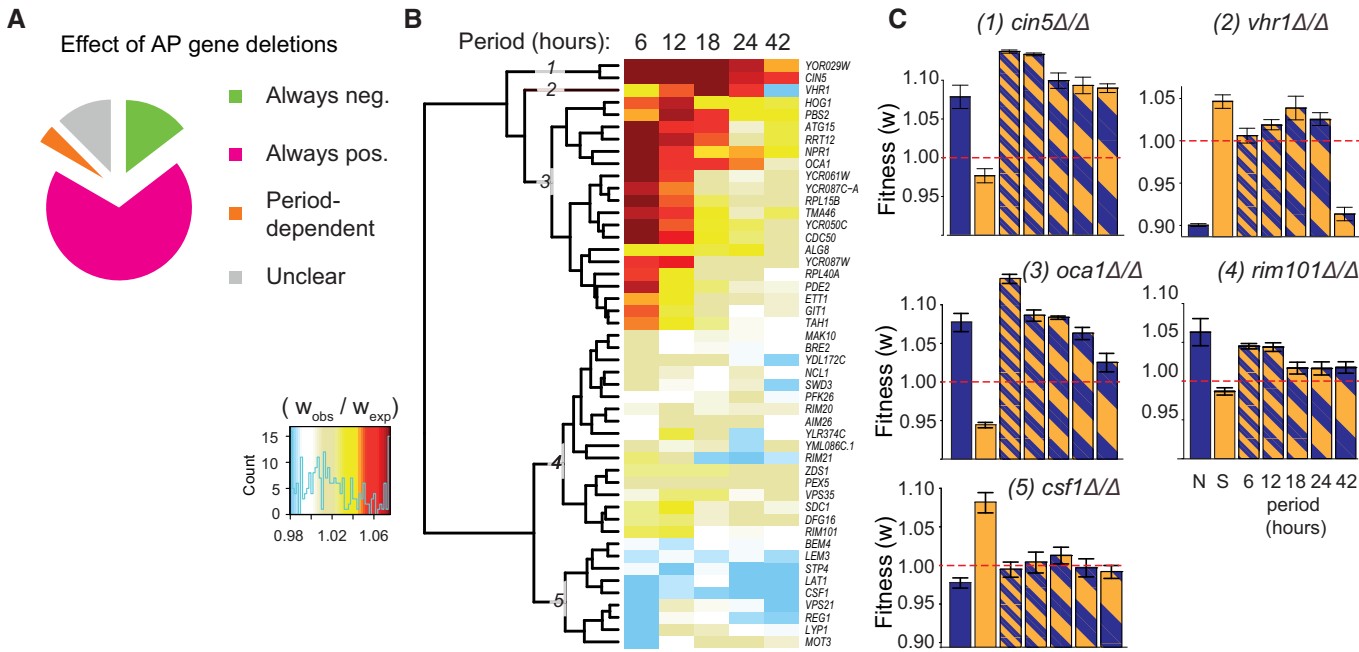

**Figure 4. Oscillations between antagonistic conditions.**

A The 48 gene deletions with significant antagonistic pleiotropy (AP) between N and S were classified according to their direction of effect on growth in periodic salt stress ("pos"itive = advantageous, "always" = at all periods of oscillations, "neg"ative = disadvantageous).

B Hierarchical clustering of AP deletions according to fitness inhomogeneity in periodic salt stress.

C Fitness values of five mutants representative of the clusters shown in (B). Bars: mean ± SEM, $n$ = 3 (N) or 4 (others) replicate cultures.

By using the available replicate fitness values, we detected 55 gene deletions where fitness in the 6-h periodic stress was significantly higher than the maximum of fitness in N and in S and 23 gene deletions where it was lower than the minimum (Fig 5A, FDR = 0.03, see Materials and Methods). Importantly, transgressivity was observed not only from BAR-Seq but also when studying gene deletions one by one in competition assays, as shown for *pde2Δ/Δ, tom7Δ/Δ, trm1Δ/Δ* and *yjl135wΔ/Δ* (Fig 5B–E). This transgressivity may have important implications on the spectrum of mutations found in hyperproliferative clones that experienced repetitive stress (see Discussion). Strikingly, the gene deletions displaying this effect were associated with various cellular and molecular processes: cAMP/PKA (*pde2Δ/Δ*), protein import into mitochondria (*tom7Δ/Δ*), autophagy (*atg15Δ/Δ*), tRNA modification (*trm1Δ/Δ*), phosphatidylcholine hydrolysis (*srf1Δ/Δ*) and MAPK signalling (*ssk1Δ/Δ, ssk2Δ/Δ*); and some of these molecular functions were not previously associated with salt stress.

**The high-affinity cAMP phosphodiesterase and Tom7p are necessary to limit hyperproliferation during periodic salt stress**

As mentioned above, several gene deletions impairing the cAMP/PKA pathway displayed inhomogeneous fitness under salt oscillations (Appendix Table S2). One of them, *pde2Δ/Δ,* had a particularly marked fitness transgressivity (Fig 5B). To determine whether this effect truly resulted from the loss of PDE2 activity, and not from secondary mutations or perturbed regulations of neighbouring genes at the locus, we performed a complementation assay. Re-inserting a wild-type copy of the gene at another genomic locus

reduced hyperproliferation and fully abolished fitness transgressivity (Fig 5F). Thus, the observed effect of *pde2Δ/Δ* directly results from the loss of Pde2p, the high-affinity phosphodiesterase that converts cAMP to AMP (Wilson & Tatchell, 1988), showing that proper cAMP levels are needed to control proliferation during repeated salinity changes.

Unexpectedly, we found that deletion of TOM7, which has so far not been associated to saline stress, also caused fitness transgressivity in the 6-h periodic regime (Fig 5C). The Tom7p protein regulates the biogenesis dynamics of the translocase of outer membrane (TOM) complex, the major entry gate of cytosolic proteins into mitochondria (Neupert & Herrmann, 2007), by affecting both the maturation of the central protein Tom40p and the later addition of Tom22p (Yamano *et al*, 2010; Becker *et al*, 2011). We observed that re-inserting a single copy of TOM7 in the homozygous diploid mutant was enough to reduce hyperproliferation, although not to the levels of the wild-type diploid, and abolished fitness transgressivity (Fig 5G). This suggests that proper dynamics of TOM assembly at the outer mitochondrial membrane are needed to limit proliferation during salinity oscillations.

**Specificity vs. pleiotropy of fitness inhomogeneity**

The fitness properties described above may be specific to salt stress, or they may be general to environmental periodicity. We therefore asked whether a periodic environment unrelated to salt stress would also favour the proliferation of numerous gene deletion mutants, and whether a common sense-and-respond pathway might be

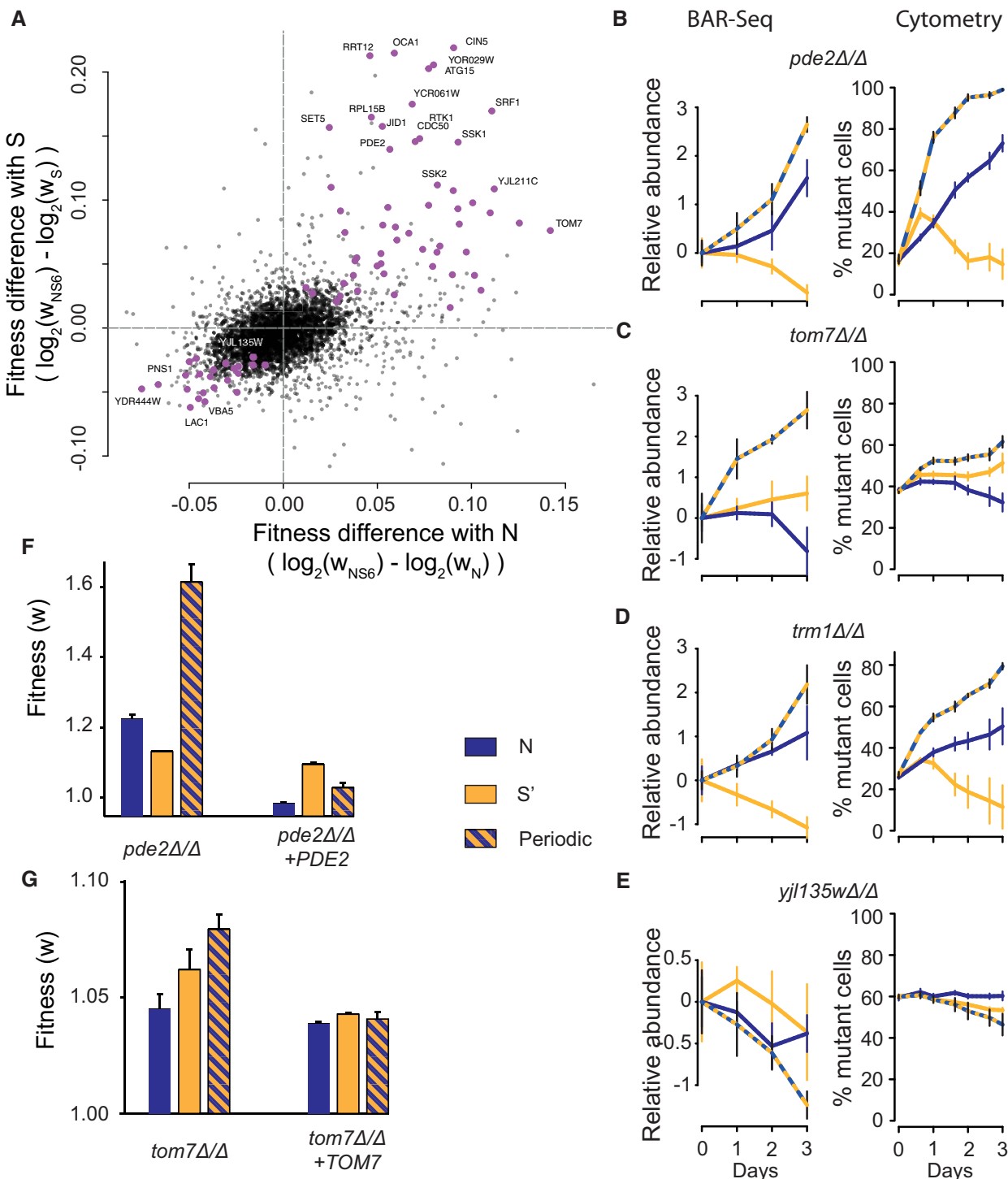

**Figure 5. Extreme proliferation rates emerging from salinity oscillations.**

A       Scatterplot of all mutants showing their observed fitness in the 6-h periodic salt stress (NS6) relative to their fitness in N (*x*-axis) and S (*y*-axis). Violet, 78 mutants with significant transgressivity (FDR = 0.03).

B–E    Time course of mutant abundance in the pool of all mutants (BAR-Seq, left, as in Fig 1B) or when the mutant was individually co-cultured with GFP-tagged wild-type cells (flow cytometry, right, as in Fig 1G). Median values ± SD (*n* = 4 replicate cultures, except for BAR-Seq N condition at day 3: *n* = 3). Conditions: N (blue), S (yellow), NS6 (hatching).

F–G    Complementation assays. Diploid homozygous deletion mutants for *pde2* and *tom7* (strains GY1821 and GY1804, respectively) were complemented by integration of the wild-type gene at the *HO* locus (strains GY1929 and GY1921, respectively). Strains were co-cultured for 24 h with GFP-tagged wild-type cells (strain GY1961), and relative fitness was measured by flow cytometry. Conditions: N (blue), S′ (0.4 M NaCl; orange) and 6-h periodic oscillations between N and S′ (hatching). Bars, mean fitness ± SEM (*n* = 3 replicate cultures).

                                

associated with higher-than-expected fitness in unrelated periodic environments.

We addressed these questions by repeating our genomic experiment in an environment that alternated between full activation and full repression of methionine biosynthesis (0 and 1 mM extracellular methionine, respectively) (Thomas & Surdin-Kerjan, 1997). We observed that with 6-h oscillations, fitness

inhomogeneity was frequent. We were able to identify 217 genes for which inhomogeneity was highly significant (Appendix Table S1, FDR = 0.0001). The associated genomic pattern was, however, notably different from the one observed in periodic salt stress. The extent of inhomogeneity was less pronounced, and the overall direction of inhomogeneity was towards a reduction, rather than an increase, in fitness (Fig 6A). Interestingly, we

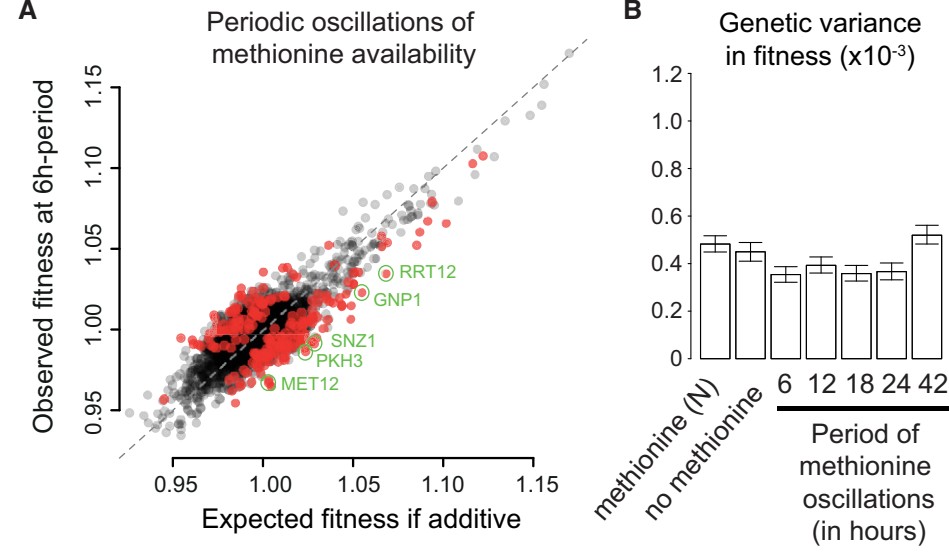

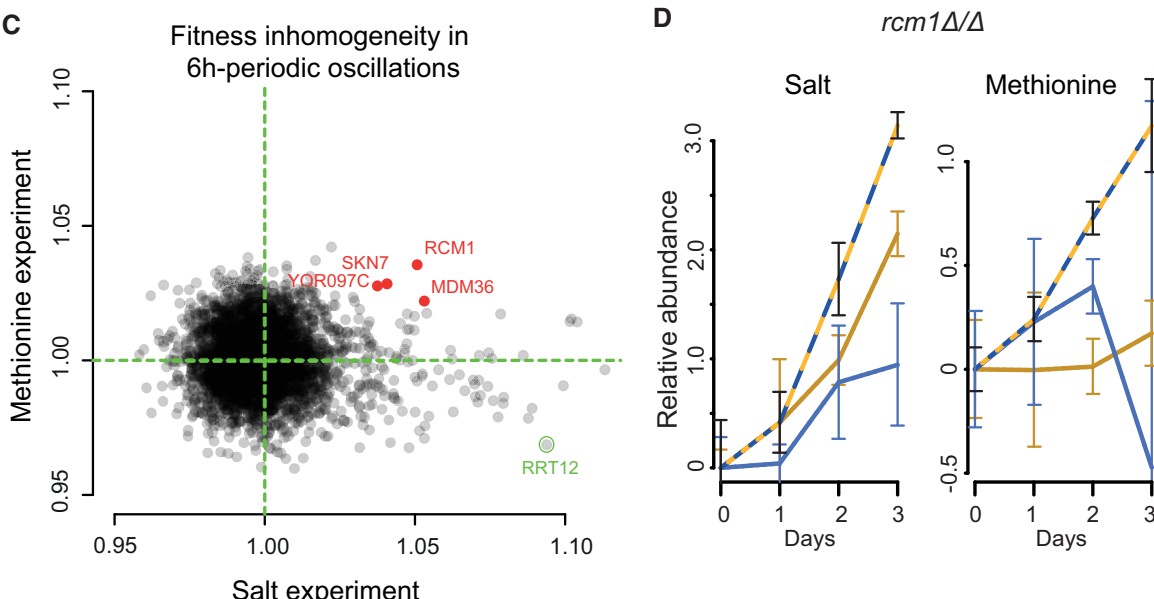

**Figure 6. Genomics of fitness in periodic fluctuations of extracellular methionine.**

A  Observed vs. expected fitness at 6-h periodic fluctuations (same representation as Fig 1E). Red dots: 217 mutants with significant inhomogeneity (FDR = 0.0001).
B  Genetic variance in fitness of the pooled population of mutants for each condition. The same *y*-axis scale as Fig 2C was used to allow comparison. Bars: 95% CI bootstrap intervals. Values are median ± SD; number of replicates: M1 day 2: *n* = 3, M1 day 3: *n* = 2, M0 day 2 and day 3: *n* = 3, N day 3: *n* = 3, all others: *n* = 4.
C  Fitness inhomogeneity ($w_{obs}/w_{exp}$) in periodic fluctuations of methionine (*y*-axis) or salt (*x*-axis) concentrations.
D  BAR-Seq time-course data (as in Fig 1B) of *rcm1Δ/Δ* in the two experiments. Conditions: N (salt experiment, blue), S (salt experiment, yellow), M1 (methionine experiment, blue), M0 (methionine experiment, yellow), 6-h periodic oscillations (hatching). Mutant abundance in condition M1 at day 3 is unclear (very large error bar).

recovered one mutant (*rrt12Δ/Δ*) having fitness increased by oscillations in salt, but reduced by oscillations in methionine (Fig 6C). Consistent with a weaker overall inhomogeneity, the genetic variance of the pooled population of mutants was not enlarged by methionine oscillations (Fig 6B). At the genomic scale, fitness inhomogeneities caused by methionine and salt oscillations were not correlated (Fig 6C). We could, however, identify a handful of mutants displaying higher-than-expected fitness in the two regimes. One of them, *rcm1Δ/Δ* (Fig 6D), corresponds to a 25S rRNA methyltransferase homologous to human NSUN5, which is deleted in Williams–Beuren Syndrome. The fitness advantage of this mutant in both salt and methionine oscillations might result from its general stress resistance conferred by widespread translational reprogramming (Schosserer *et al*, 2015) and from variations in intracellular levels of the methyl donor AdoMet when extracellular concentrations of methionine fluctuate (Thomas & Surdin-Kerjan, 1997).

In conclusion, the proliferative advantage of numerous gene deletion mutants at short periods of environmental changes is specific to salt stress, and fitness inhomogeneity of most mutants is specific to the oscillating environment.

# Discussion

We quantified the contribution of 3,568 yeast genes to cell growth during periodic salt stress and during periodic methionine availability. These surveys identified 456 and 217 genes, respectively, for which fitness did not match the time average of the fitness in the two alternating conditions. This widespread (and sometimes extreme) time inhomogeneity of the genetic control of cell proliferation has several important implications.

## Novel information is obtained when studying adaptation out of equilibrium

A large part of the information about the properties of a responsive system is hidden at steady state. For example, a high protein level does not distinguish between fast production and slow degradation. For this reason, engineers working on control theory commonly study complex systems by applying periodic stimulations, as a way to explore the system's behaviour out of equilibrium. In particular, spectral analysis can sometimes demonstrate vulnerabilities of biological systems (Ang *et al*, 2011; Mitchell *et al*, 2015).

In the present study, a genomic screen under periodic stress revealed two features of the salt stress response that were not suspected. Numerous gene deletions exacerbate hyperproliferation at short alternating periods (Figs 2A and 5A); and many of the concerned genes were not previously associated with salt stress (e.g. TOM7, ATG15, SRF1, RPL15B, RRT12). Thus, dynamic and repeated stimulations can reveal hidden properties of a well-known biological system.

## Gene × Environment interactions in dynamic conditions

Interactions between genes and environmental factors (G × E) are omnipresent and constitute the driving force for the adaptation of populations. Our results showed that, for many mutations, G × E

may not be predictable under certain environmental dynamics. The way G × E is affected by environmental dynamics probably differs between a periodic stress that is natural to an organism and a periodic stress that has never been experienced by the population (as considered here). In the first case, populations can evolve molecular clocks adapted to the stress period. Nature is full of examples, artificial clocks can be obtained by experimental evolution of microorganisms (Wildenberg & Murray, 2014), and nematodes evolving under anoxia/normoxia transitions at each generation were reported to evolve an anticipatory mechanism: hermaphrodites produced more glycogen during normoxia, thereby protecting their eggs to the upcoming anoxia at the expense of glycerol that they themselves needed (Dey *et al*, 2016). In contrast, when a periodic stress is encountered for the first time, cells face a novel challenge. The dynamic properties of their stress response can then generate extreme phenotypes, such as the hyperproliferation described here (Figs 2B and 5A–D), or long-term growth arrest as described by others (Mitchell *et al*, 2015).

## Natural selection in dynamic environments

Because the traits we quantified were the relative rates of proliferation between different genotypes (fitness), our survey provides a genome-scale view of the selection of mutations in periodic environments. Theoretical studies have shown that complex interactions between the dynamics of the environment and the dynamics of adaptation (population size, allele frequencies or target size for beneficial mutations) can affect selection (Cvijović *et al*, 2015; Sæther & Engen, 2015; Svardal *et al*, 2015). For example, simulations by Cvijovic *et al* evidenced a reduced selection of *de novo* mutations appearing during slow environmental oscillations with seasonal drift (Cvijović *et al*, 2015). We report here the emergence of strong positive selection on pre-existing mutations when novel, fast and strictly periodic salinity oscillations occur. Different types of inhomogeneity may therefore participate to the complexity of selection in natural environments. In particular, our observation of several cases of transgressive fitness suggests that environmental oscillations on short timescales can twist natural selection in favour of a subset of mutations on the long term.

We observed that the diversity of fitness among the pooled population of mutants could be modified by environmental dynamics: the shorter the period of salt stress, the stronger were the differences. This finding is important because, according to Fisher's theorem, genetic variation in fitness reflects the rate of population adaptation (Fisher, 1958; Frank & Slatkin, 1992). Our observations therefore directly couple two timescales: fast dynamics at the level of environmental oscillations, with long-term changes of the population. Several early studies of experimental evolution of *Drosophila* flies in either steady or fluctuating conditions showed that the genetic variance of fitness-related traits increased in fluctuating regimes (Beardmore, 1961; Mackay, 1979; Verdonck, 1987). In our study, the genetic diversity (a large pool of *de novo* mutations) pre-existed the fluctuations and the observed elevated genetic variance in fitness corresponds to a large diversity of selection coefficients (fitness itself) acting on the mutations when the environment oscillates. Thus, both the genetic variance of fitness-related traits and the diversity of the selection coefficients likely participate to the coupling between the short timescales of

environmental fluctuations and the long timescales of population adaptation.

### Stress amnesia of chromatin modifying mutants?

Yeast cells are known to "record" stress occurrence via molecular changes conferring long-term (epigenetic) memory associated with an improved response at later exposures (Guan *et al*, 2012). In the case of salt stress, this process involves chromatin modifications mediated by the Set1/COMPASS complex (D'Urso *et al*, 2016). Mutants of this complex displayed a systematic fitness pattern in our data. Removal of either one of five components (Swd1p, Spp1p, Sdc1p, Swd3p and Bre2p) decreased fitness in N, increased it in S and increased it similarly in the periodic regime (Appendix Fig S4). We also observed abundant fitness inhomogeneities for numerous mutants of other chromatin modifying complexes, such as *rtt106Δ/Δ*, *set5Δ/Δ*, *swr1Δ/Δ*, *vps72Δ/Δ*, *hst3Δ/Δ* or *cac2Δ/Δ* (Dataset EV1). This could result from memory alterations that change the response dynamics in ways that are better suited to the periodic regime. Alternatively, it could result from a trade-off: memorization consumes energy (remodelling) and chemicals (e.g. AdoMet), and modifies chromatin instead of leaving it free to replicate. When occurring repeatedly, this likely penalizes the growth of wild-type cells as compared to mutants where the process is defective. Our screen opens the possibility to further investigate the contribution of specific chromatin factors to stress memorization, for example by tracking the dynamics of growth of individual mutant cells in a controlled dynamic environment (Mitchell *et al*, 2015; Llamosi *et al*, 2016).

### The high-affinity cAMP phosphodiesterase constitutes a genetic vulnerability to environmental dynamics

One of the mutants unexpectedly fit for periodic salt stress was *pde2Δ/Δ*, and this phenotype was complemented by ectopic re-insertion of a wild-type copy of the gene. The yeast genome encodes two phosphodiesterases: one of low affinity that shares homology with only a fraction of eukaryotes (Pde1p) and one of high affinity that belongs to a well-studied class of phosphodiesterases found in many species, including mammals (Pde2p) (Ma *et al*, 1999). Our genomic data did not indicate any obvious fitness alteration of *pde1Δ/Δ* cells in alternating conditions (Dataset EV1). These two enzymes convert cAMP into AMP, thereby reducing protein kinase A activity which is implicated in response to various stress including high salt (Norbeck & Blomberg, 2000; Park *et al*, 2005). Negative regulators of the pathway, including *PDE2*, are recurrent targets for *de novo* mutations in yeast populations evolving in serial transfer experiments (Venkataram *et al*, 2016) and for natural standing variation affecting proliferation under stressful conditions (Parts *et al*, 2011). The fitness transgressivity of *pde2Δ/Δ* cells in periodic salt stress suggests that the positive selection of such mutations may be even stronger if environmental conditions fluctuate. In addition, the output of the cAMP/PKA pathway is likely governed by its dynamic properties, since intracellular levels of cAMP oscillate (Gonzales *et al*, 2013), with consequences on nucleo-cytoplasmic oscillations of Msn2p (Garmendia-Torres *et al*, 2007). The activity of Pde2p is itself modulated by PKA (Hu *et al*, 2010, 2), and this negative feedback is probably important for suitable dynamics (Gonzales *et al*, 2013). Our results suggest that loss of this feedback confers a hyperproliferative advantage and that it therefore constitutes a genetic vulnerability during prolonged exposure to periodic stress.

### Distinct genomic profiles between salt and methionine oscillations

The pronounced fitness alterations seen under repeated salt stress were not observed under repeated variation of extracellular methionine. This difference could be due to different target sizes in the genome. The effect of high-/low-salt transitions is global, affecting osmolarity, cell size, membranes and ionic contents in general. The number of genes that may couple the dynamics of transitions to cell growth in this context is large. In contrast, transitions in the presence/absence of methionine are not generally stressful but affect a specific metabolic process covered by fewer genes.

In addition, the two experiments revealed different directions of fitness inhomogeneity: salt oscillations increased fitness of many deletion mutants, whereas methionine oscillations tended to penalize fitness of several mutants. This is consistent with a higher cost of triggering an unnecessary cell response for the global cellular changes following salt stress than for the specific metabolic changes following methionine deprivation.

### Relevance to cancer

Cancer is an evolutionary issue: hyperproliferative cells possessing tumorigenic somatic mutations accumulate in tissues and threaten the whole body's life. This process is driven by both the occurrence of these mutations (mutational input) and the natural selection of somatic mutations among cells of the body. The effect of mutations on proliferation rates is a central component of the process of selection. Human tissues are paced by various dynamics. Some rhythms are natural (sleep, food intake, hormonal cycles, circadian clocks, walking steps, etc.), and others are artificial (mechanic and electromagnetic waves, periodic medicine intake). The impact of these dynamics on the selection process of somatic mutations is unknown. Our results on yeast suggest that it may be significant, because a transient episode of periodic stress may strongly reshape allele frequencies in a population of mutant cells. Importantly, the molecular processes that are exposed to this phenomenon are common to all eukaryotes (cAMP/PKA, autophagy, tRNA modifications, protein import in mitochondria). Now that barcoding techniques allow researchers to track selection in cancer cell lines (Bhang *et al*, 2015); using them in a context of periodic stimulations may reveal unsuspected genetic factors.

## Materials and Methods

### Yeast deletion library and growth media

The pooled homozygous diploid yeast deletion library was purchased from Invitrogen (ref. 95401.H1Pool). In each strain, the coding sequence of one gene had been replaced by a KanMX4 cassette and two unique barcodes (uptag and downtag) flanked by universal primers (Winzeler *et al*, 1999). Following delivery, the

yeast pool was grown overnight in 100 ml YPD medium, and 500 µl aliquots ($2.2 \times 10^8$ cells/ml) were stored in 25% glycerol at −80°C. For salt stress oscillations, medium N (Normal) was a synthetic complete medium made of 20 g/l D-glucose, 6.7 g/l yeast nitrogen base without amino acids (Difco), 88.9 mg/l uracil, 44.4 mg/l adenine, 177.8 mg/l leucine and all other amino acids at 88.9 mg/l and 170 µl/l NaOH 10 N. Medium S (Salt) was made by adding 40 ml/l NaCl 5 M to medium N (final concentration of 0.2 M). For methionine oscillations, medium M0 was a synthetic medium similar to N but lacking methionine, and medium M1 was made by supplementing M0 with 149 mg/l (1 mM) methionine.

### Experimental set-up for environmental oscillations

All steps of the oscillation experiment were carried out in 96-well sterile microplates using a Freedom EVO200 liquid handler (Tecan) equipped with a 96-channel pipetting head (MCA), a high precision 8-channel pipetting arm (LiHa), a robotic manipulator arm (RoMa), a Sunrise plate reader (Tecan), a MOI-6 incubator (Tecan) and a vacuum station (Millipore). All robotic steps were programmed in Evoware v2.5.4.0 (Tecan). The salt experiment included seven culture conditions (N, S, NS6, NS12, NS18, NS24 and NS42), where N and S were steady environments, and NSx was an environment alternating between N and S with a period of x hours. The methionine experiment also included seven culture conditions (steady M0; steady M1; and alternating conditions with periods of 6, 12, 18, 24 or 42 h). Each condition was applied on four independent populations. To reduce technical variability and population bottlenecks, each population was dispatched in four parallel microplates before each incubation step and these plates were combined into a single one after incubation. The size of each population was maintained over $2.1 \times 10^7$ cells.

### Initialization of pooled-mutant cultures

Four aliquots of the yeast deletion library were thawed, pooled and immediately diluted into 100 ml of fresh N medium. After mixing, samples of 220 µl of the cell suspension were immediately distributed into 28 wells of each of four distinct microplates. This initiated a total of 112 populations of cells, each containing ~320 copies of each mutant strain on average. Plates were then incubated at 30°C for 6 h with 270 rpm shaking.

### Environmental oscillations of pooled-mutant cultures

Twice a day, a stock of source plates that contained sterile N or S fresh medium in the appropriate wells was prepared. Every 3 h, the four microplates containing cells were removed from the incubator (30°C, 270 rpm) and cells were transferred to a single sterile plate having a 1.2-µm-pore filter bottom (Millipore, MSBVS1210). Media were removed by aspiration, and four fresh source plates were extracted from the stock. 62 µl of sterile media was pipetted from each source plate and transferred to the filter plate, cells were resuspended by pipetting 220 µl up and down, and 60 µl of cell suspension was transferred to each of the four source plates which were then incubated at 30°C with 270 rpm for another 3 h. Every 6 h, cell density was monitored for one of the four replicate plates by $OD_{600}$ absorbance. Every 24 h, 120 µl of cultures from each replicate plate

was sampled, pooled in a single microplate and centrifuged 10 min at 5,000 g; and cell pellets were frozen at −80°C. Dilution rates of the populations were as follows: 85% when the action was only to replace the media, 55% when it was to replace the media and to measure OD, and 32% when it was to replace the media, to measure OD and to store samples.

The experiment lasted 78 h in total and generated samples from 28 independent populations at time points 6 h (end of initialization), 30, 54 and 78 h.

### BAR-Seq

Frozen yeast pellets were resuspended in 200 µl of a mix of 30 ml of Y1 Buffer (91.1 g of sorbitol in 300 ml $H_2O$, 100 ml of 0.5 M EDTA, 0.5 ml of β-mercaptoethanol, completed with 500 ml of water), 60 units of zymolyase (MP Biomedicals, ref 8320921) and 22.5 µl of RNAse A at 34 mg/ml (Sigma ref R4642), vortexed and incubated for 1 h at 37°C for cell wall digestion. Genomic DNA (gDNA) was extracted by using the Macherey Nagel 96-well NucleoSpin Kit (ref 740741.24) following manufacturer's instructions. We designed and ordered from Eurogentec a set of 112 reverse primers of the form 5′-P5-$X_9$-U2-3′, where P5 (5′-AATGAT ACGGCGACCACCGAGATCTACACTCTTTCCCTACACGACGCTCTTC CGATCT-3′) allowed Illumina sequencing, $X_9$ was a custom index of nine nucleotides allowing multiplexing via a Hamming code (Bystrykh, 2012), and U2 (5′-GTCGACCTGCAGCGTACG-3′) matched a universal tag located downstream of the uptag barcode of each mutant yeast strain. PCR amplification of the barcodes of each sample was done by using these reverse primers in combination with one forward primer of the form 5′-P7-U1-3′, where P7 (5′-CAAGCAGAAGACGGCATACGAGATGTGACTGGAGTTCAGACGTGT GCTCTTCCGATCT-3′) allowed Illumina sequencing and U1 (5′-GATGTCCACGAGGTCTCT-3′) matched a universal tag located upstream of the uptag barcode of each yeast mutant. Reagents used for one PCR were as follows: 18.3 µl of water, 6 µl of buffer HF 5× and 0.2 µl of Phusion polymerase (ThermoFischer Scientific, ref F530-L), 2.5 µl of dNTP 2.5 mM, 1 µl of each primer at 333 nM and 1 µl of gDNA at 300 to 400 ng/µl. Annealing temperature was 52°C, extension time was 30 s, and 30 cycles were performed. As observed previously, the PCR product migrated as two bands on agarose gels, which can be explained by heteroduplexes (Pierce et al, 2007). Both bands were extracted from the gel, purified and eluted in 30 µl water. All 112 amplification products were pooled together (10 µl of each), gel-purified and eluted in a final volume of 30 µl water and sequenced by 50 nt single reads on a Illumina HiSeq2500 sequencer by ViroScan3D/ProfileXpert (Lyon, France).

### Data extraction, filtering and normalization

Demultiplexing was done via an error-correction Hamming code as described previously (Bystrykh, 2012). Mapping (assignment of reads to yeast mutant barcodes) was done by allowing a maximal Levenstein distance of 1 between a read and any sequence in the corrected list of mutant barcodes of Smith et al (2009). For the salt experiment, a total of 291 million reads were mapped and used to build a raw 6,004 (mutants) by 112 (samples) table of counts. One sample was discarded because it was covered by < 300,000 total

counts and displayed mutant frequencies that were poorly correlated with their relevant replicates. Similarly, 2,436 mutants were covered by few (< 2,000) counts over all samples (including samples of another unrelated experiment that was sequenced in parallel) and were discarded, leaving a table of 153,908,522 counts, corresponding to 3,568 mutants in 111 samples for further analysis. Sequence reads from the methionine experiments were mapped and used similarly to build another table of 25,184,773 counts corresponding to 3,568 mutants in 98 samples. For the methionine experiment, the four samples of day 0 corresponding to steady M1 condition had been replaced by technical controls. The corresponding missing data of the final count table were replaced by the median counts of 11 other populations at day 0 (which also grew in N medium during initialization). These tables were then separately normalized using the function *variance-StabilizingTransformation* from the DESeq2 package (Love *et al*, 2014) (version 1.8.1) with arguments blind = FALSE and fitType = "local".

## Fitness estimation

We followed the method of Qian *et al* (2012) to estimate the fitness cost or gain ($w$) of each mutant in each population. Eleven genes (Appendix Table S4) were considered to be pseudogenes or genes with no effect on growth, and the data from the corresponding deletion mutants were combined and used as an artificial "wild-type" reference. For each mutant strain $M$, $w$ was calculated as follows:

$$w = \left( \frac{M_e/M_b}{WT_e/WT_b} \right)^{1/g}$$

with $M_b$, $M_e$, $WT_b$ and $WT_e$ being the frequencies of strain $M$ and artificial wild-type ($WT$) strain at the beginning ($b$) or end ($e$) of the experiment, and $g$ the number of generations in between. $g$ was estimated from optical densities at 600 nm of the entire population. We fixed $g = 24$ (eight generations per day, doubling time of 3 h) although doubling time was ~10% longer in steady S than in steady N. Accounting for this difference in the estimation of $w_S$ did not affect significantly the estimation of fitness inhomogeneity (Appendix Fig S5). Note that alternative estimates of fitness have been proposed that account for possible fitness variation over the course of the experiment (Schlecht *et al*, 2017).

## Deviation from time-average fitness

We analysed fitness inhomogeneity by both quantifying it and testing against the null hypothesis of additivity. The quantification was done by computing $dev = w_{observed}/w_{expected}$, where $w_{observed}$ was the fitness of the mutant strain experimentally measured in the periodic environment and $w_{expected}$ was the fitness expected given the fitness of the mutant strain in the two steady environments (N and S), calculated

$$w_{expected} = w_N{}^{f_N} \cdot w_S{}^{f_S}$$

with $f_N$ and $f_S$ being the fraction of time spent in N and S media, respectively, during the course of the oscillation experiment

($f_N = f_S = 0.5$ for all periods except 42 h). Statistical inference was based on a generalized linear model applied to the normalized count data (one model per oscillating period). We assumed that the normalized counts of mutant $i$ in condition $c$ (N, S or periodic) at day $d$ in replicate population $r$ originated from a negative binomial distribution NB($\lambda_i$, $\alpha$), with:

$$\log(\lambda_i) = \text{offset}_{i,c} + \beta_{i,1} \cdot t_{c,d}^N + \beta_{i,2} \cdot t_{c,d}^S + \beta_{i,3} \cdot N_{c,d}^{changes} + \varepsilon_{i,c,d,r}$$

and offset$_{i,c}$ being the median of normalized counts for condition $c$ at day 0, $t_{c,d}^N$ and $t_{c,d}^S$ being the amount of time spent in medium N and medium S at day $d$, respectively ($d/2$ in most cases), $N_{c,d}^{changes}$ being the number of changes between the two media that took place between days 0 and $d$, and $\varepsilon$ being the residual error. The model was implemented in R using the function *glm.nb* of the MASS package (version 7.3-40).

If fitness is homogenized in a oscillating environment, then it is insensitive to the number of changes and $\beta_{i,3} = 0$. Inhomogeneity can therefore be inferred from the statistical significance of the term $N_{c,d}^{changes}$ of the model. The corresponding *P*-values were converted to *q*-values, using package *qvalue* version 2.0.0 in order to control the false discovery rate. All computations were the same for the methionine experiment, with N indices corresponding to the M1 medium and S indices corresponding to the M0 medium.

Genetic variance in fitness was computed for each condition as follows:

$$V_G = V_T - V_E$$

where

$$V_T = \frac{1}{3N} \sum_{i=1}^{N} \sum_{j=1}^{3} (w_{i,j} - \bar{w})^2$$

was the total variance in fitness, and

$$V_E = \frac{1}{3N} \sum_{i=1}^{N} \sum_{j=1}^{3} (w_{i,j} - \bar{w}_i)^2$$

was an estimate of the non-genetic variance in fitness (inter-replicates variability or residual variance). This term is sometimes called the "environmental variance" but we avoid this denomination here because the environment varies. $N$ was the number of gene deletions, $w_{i,j}$ the fitness of gene deletion $i$ in replicate $j$, $\bar{w}_i$ the mean fitness of gene deletion $i$, and $\bar{w}$ the global mean fitness. The 95% confidence intervals of $V_G$ were computed from 1,000 bootstrap samples (randomly picking mutant strains, with replacement).

## Antagonistic pleiotropy

We used the observed $w_N$ and $w_S$ values (fitness in the N and S steady conditions, respectively) of the deletion mutants to determine whether a mutation was antagonistically pleiotropic (AP). Our experiment provided, for each mutant, three independent estimates of $w_N$ and four independent estimates of $w_S$ (replicate populations). For each mutant, we combined these estimates in three pairs of ($w_N$, $w_S$) values by randomly discarding one of the four available $w_S$

values, and these pairs were considered as three independent observations. We considered that an observation supported AP if the fitness values ($w_N$, $w_S$) showed (1) an advantage in one of the conditions and a disadvantage in the other and (2) deviation from the distribution of observed values in all mutants, since most deletions are not supposed to be AP. Condition (1) corresponded to ($w_N > 1$ AND $w_S < 1$) OR ($w_N < 1$ AND $w_S > 1$). Condition (2) was tested by fitting a bivariate Gaussian to all observed ($w_N$, $w_S$) pairs and labelling those falling two standard deviations away from the model (Appendix Fig S3). A deletion was considered AP if all replicates (three observations) supported AP, which was the case for 48 deletions. A permutation test (re-assigning observations to different deletions replicates) determined that less than one deletion (0.54 on average) was expected to have three observations supporting AP by chance only (Appendix Table S3). For the selected 48 deletions, the magnitude of AP was computed as $w_N / w_S$. For each deletion, the direction of selection (Fig 4A) in each condition was considered to be positive if $\bar{w} - \sigma_w > 1$ , negative if $\bar{w} + \sigma_w < 1$ and ambiguous otherwise, with $\bar{w}$ and $\sigma_w$ being the mean and standard deviation of fitness values across replicates, respectively. A mutation was classified as follow: "unclear" if its direction of selection was ambiguous at four or five oscillating periods, "always positive" if all its unambiguous directions of selection were positive, "always negative" if all its unambiguous directions of selection were negative, and "period dependent" if its direction differed between periods.

### Transgressive fitness

We considered that a mutant had transgressive fitness if at least three of its four observed replicate measures of fitness in oscillating conditions ($w_{NS}$) were either all higher than $\max(\bar{w}_N + \sigma_N, \bar{w}_S + \sigma_S)$ or all lower than $\min(\bar{w}_N - \sigma_N, \bar{w}_S - \sigma_S)$, where $\bar{w}_N$ (respectively $\bar{w}_S$) was the mean fitness value in steady condition N (respectively S), and $\sigma_N$ (respectively $\sigma_S$) the corresponding standard deviation. A permutation test (re-assigning observations to random mutants) determined that less than three mutants (2.24 on average) were expected to display three replicates supporting transgressivity by chance only (Appendix Table S5).

### Direct fitness measurement by flow cytometry: plasmids and strains

Individual homozygous diploid knock-out strains and the control wild-type strain BY4743 were ordered from Euroscarf. Oligonucleotides and modified strains used in this study are listed in Appendix Tables S6 and S7, respectively. We constructed a GFP-tagged wild-type strain (GY1738), and its non-GFP control (GY1735), by transforming BY4743 with plasmids pGY248 and HO-poly-KanMX4-HO (Voth *et al*, 2001), respectively. Plasmid pGY248 was ordered from GeneCust who synthesized a Pact1-yEGFP BamHI fragment and cloned it into HO-poly-KanMX4-HO. Complemented strains were generated by cloning the wild-type copy of each gene of interest into a plasmid targeting integration at the *HO* locus. We first prepared a vector (pGY434) by removing the repeated *hisG* sequence of plasmid HO-hisG-URA3-hisG-poly-HO (Voth *et al*, 2001) by SmaI digestion and religation followed by ClaI digestion and religation. For *PDE2*, the wild-type (S288c) coding sequence with its 600-bp upstream and 400-bp downstream regions was

synthesized by GeneCust and cloned in the BglII site of pGY434. The resulting plasmid (pGY453) was digested with NotI and transformed in strain GY1821 to give GY1929. For *TOM7*, we constructed plasmid pGY438 by amplifying the HOL-URA3-HOR fragment of pGY434 with primers 1O21 and 1O22, and cloning it into pRS315 (Brachmann *et al*, 1998) (linearized at NotI) by *in vivo* recombination. The wild-type copy of *TOM7* (coding sequence with its 465-bp upstream and 813-bp downstream regions) was PCR-amplified from strain BY4742 using primers 1O27 and 1O28 and co-transformed in BY4742 with PacI-PmeI fragment of pGY438 for *in vivo* recombination. The resulting plasmid (pGY442) was digested by NotI and the 4-kb fragment containing HO-URA3-TOM7-HO was gel-purified and transformed in GY1804 to obtain GY1921. Proper integration at the HO locus was verified by PCR. Since complementation was accompanied by the URA3 marker, which likely contributes to fitness, we competed strains GY1921 and GY1929 with a URA$^+$ wild-type strain (GY1961), which was obtained by transforming strain GY1738 with the PCR-amplified URA3 gene of BY4716 (with primers 1D11 and 1D12). The non-GFP control URA$^+$ wild-type strain GY1958 was obtained similarly.

### Direct fitness measurement by flow cytometry: oscillating cultures

Each plate contained eight different mixed cultures (one per row) and three different conditions (N, S, NS6) with four replicates each that were randomized (neighbouring columns contained different conditions). Four plates were handled in parallel, which allowed us to test 32 different co-cultures per run, with at least one row per plate dedicated to controls (wild-type strain vs. itself or wild-type strain alone). Strains were streaked on G418-containing plates. Single colonies were used to inoculate 5 ml of N medium and were grown overnight at 30°C with 220 rpm shaking. The next day, concentration of each culture was adjusted to an OD$_{600}$ of 0.2. For co-cultures, 2 ml of wild-type cell suspension was mixed with 2 ml of mutant cell suspension, and 220 μl of this mix was transferred to the desired wells of a microplate. Plates were then incubated on the robotic platform at 30°C with 270 rpm for 4–5 h. Oscillations of the medium condition were also done by robotics: dilution (keeping 130 μl of the 220 μl cell suspension), filtration and re-fill every 3 h, using a stock of fresh source plates prepared in advance. Twice a day, 90 μl of the cell suspension was fixed and processed for flow cytometry. Fixation was done on the robotic platform, by washing cells twice with PBS 1X, re-suspending them in PBS 1X + paraformaldehyde 2% and incubating at room temperature for 8 min, washing with PBS 1X, re-suspending cells in PBS + Glycine 0.1 M, incubating at room temperature for 12 min and finally washing cells with PBS 1X and re-suspending them in PBS 1X. Plates were then diluted (at 80–95%) in PBS 1X and stored at 4°C before being analysed on a FACSCalibur flow cytometer (BD Biosciences). Acquisitions were stored on 10,000 cells at a mean rate of 1,000 cells/s.

### Direct fitness measurement by flow cytometry: data analysis

Raw .fcs files were analysed using the *flowCore* package (version 1.34.3) from Bioconductor (Hahne *et al*, 2009) and custom codes. Cells of homogeneous size were dynamically gated as follows: (i) removal of samples containing < 2,000 cells, (ii) removal of events

with saturated signals (FSC, SSC or FL1 $\geq$ 1,023 or $\leq$ 0), (iii) computation of a density kernel of FSC, SSC values to define a perimeter of peak density containing 40% of events and (iv) cell gating using this perimeter, keeping > 4,000 cells. In order to classify each cell as GFP$^+$ or GFP$^-$, FL1 thresholds were determined automatically using the function *findValleys* from package *quantmod* (version 0.4-4). The relevance of these thresholds was then verified on control samples containing only one of the two strains (unimodal GFP$^+$ or GFP$^-$). After classifying GFP$^+$ (i.e. WT) and GFP$^-$ (i.e. mutant) cells, fitness values were computed as $w = ((M_e/M_b)/(WT_e/WT_b))^{1/g}$, with $M_b$, $M_e$, $WT_b$ and $WT_e$ being the frequencies of mutant strain $M$ and wild-type ($WT$) strain at the beginning ($b$) or end ($e$) of the experiment, and $g = 24$ the number of generations in between.

### Data availability

The entire dataset is provided as Dataset EV1 of this publication. The raw sequence reads of the BAR-Seq experiments are available at https://www.ncbi.nlm.nih.gov/bioproject/ under Accession Number PRJNA358207.

**Expanded View** for this article is available online.

### Acknowledgements

We thank Julien Gagneur for suggestions on normalization and generalized linear models; Arnaud Bonnaffoux, Florent Chuffart, Pascal Hersen, Abderrahman Khila, Sébastien Lemaire, Serge Pelet and Alexandre Soulard for discussions; Julien Gagneur, Steve Garvis, Jun-Yi Leu, Stephen Proulx, Mark Siegal and Henrique Teotonio for critical reading of the manuscript; Audrey Barthelaix for initial tests on the robotic platform; David Stillman for plasmids; Sandrine Mouradian and SFR Biosciences Gerland-Lyon Sud (UMS3444/US8) for access to flow cytometers and technical assistance; BioSyL Federation and Ecofect LabEx (ANR-11-LABX-0048) for inspiring scientific events; developers of R/Bioconductor and Ubuntu for their software and three anonymous reviewers for their comments. This work was supported by the European Research Council under the European Union's Seventh Framework Programme FP7/2007-2013 Grant Agreement No. 281359 and by the Fondation ARC pour la recherche sur le cancer.

### Author contributions

JS and MR set up automated cultures; JS performed the experiments and optimized automation and analysed the data; MR designed multiplexing oligonucleotides, set up BAR-Seq libraries preparations and supervised JS for the genomic experiment; JS, MR, HD-B and EF constructed strains; JS, EF and HD-B performed flow cytometry; JS, MR, and GY implemented the GLM model, interpreted results and wrote the paper; GY conceived, designed and supervised the study.

### Conflict of interest

The authors declare that they have no conflict of interest.

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
