## [Review Process File · Molecular Systems Biology]

Genomics of cellular proliferation in periodic environmental fluctuations

Jérôme Salignon, Magali Richard, Etienne Fulcrand, H el ene Duplus-Bottin and Ga el Yvert

Review timeline:

Submission date:	20 June 2017
Editorial Decision:	1 August 2017
Revision received:	14 November 2017
Editorial Decision:	16 January 2018
Revision received:	1 February 2018
Accepted:	6 February 2018

Editor: Thomas Lemberger

Transaction Report:

1st Editorial Decision

1 August 2017

Thank you again for submitting your work to Molecular Systems Biology. We have now heard back from the three referees who agreed to evaluate your manuscript. As you will see from the reports below, the referees find the topic of your study of potential interest. They raise, however, substantial concerns on your work, which should be convincingly addressed in a major revision.

Without repeating the points raised in the reports below, the major issues refer to the following:

- the three referees find the study essentially 'observational' and feel that it remains somewhat preliminary.
- reviewer #2 makes two constructive suggestions of additional analyses that could provide further insights without asking to elucidate complete mechanistic details.
- reviewer #3 also provide suggestions, in particular towards the identification of 'universal' inhomogenous gene that may delineate a core response machinery.
- the reviewers also expressed the strong view that that text should be clarified, shortened to make it both more rigorous and more accessible.

When you resubmit your manuscript, please download our CHECKLIST (<http://embopress.org/sites/default/files/Resources/EP_Author_Checklist_Master.xlsx>) and include the completed form in your submission. *Please note* that the Author Checklist will be published

 REVIEWER REPORTS

Reviewer #1:

In this work, Salignon use the yeast gene-deletion library to compare gene fitness contributions to continuous growth with and without high osmolarity versus growth through cycles of high and low osmolarity. The main contribution is that the authors identify some gene deletion strains whose fitness is different in response to cycling environments compared to the time-averaged sum of fitness in the two steady-state conditions. Some of the differential fitness effects depend on the frequency of osmotic shifts, for reasons that are not really clear.

I find the topic and the motivation interesting, but in the end I'm not sure if the results provide enough of a mechanistic advance for publication in Mol Sys Bio. Most of the results are observational and the authors highlight a few functional categories, but the underlying mechanisms are not clear.

Major points:

1. I found the manuscript difficult to read in places, including where some key methods were discussed. It was unclear which samples were submitted for sequencing - the authors describe cycling cells every 3 hours and at several points in the manuscript they discuss the 6-hour sample - but were barcodes sequenced after only 6h growth? If so, in 96-well plates, this can hardly be more than two doublings and it's hard to imagine that barcodes could be distinguished (since bar-seq experiments rely on generation times to give a real signal).
2. Since the assay is so dependent on generation times, it should be more clearly described how many generations cells passed through for each experiment - it is also fundamentally important that cells in different conditions progress through the same number of culture generations to enable comparisons, but it seems here that everything was done based on time, not generations.
3. Data in Figure 2 suggest that cells were cycled at different intervals, but I couldn't find a clear description of that in the Methods. Again it was not clear if the culture went through the same number of generations in each of these experiments, and if not it's hard to know if these are really comparable datasets.

Minor points:

4. On page 9, the authors first describe that nearly a third of the yeast gene deletion strains deviate from the time-averaged expectation, but this is at an FDR of 0.2. Most of the manuscript focuses on a more stringent list at a reasonable FDR, but then in the Discussion they refer back to this very large dataset at a relaxed FDR. Best to focus on the more stringent gene set, which is plenty of genes for analysis; there seems to be no good reason to relax the FDR cutoff.
5. Page 10: under 'Impact of environmental dynamics on mutants proliferation': this is the first the authors mention different oscillation time scales, and the text implies they are doing something computational - but data in Fig 2 looks like they did different cycling times. This should be more clearly presented in the text and the Methods.
6. Some of the terminology is confusing, e.g. this sentence on P12: "... For 33 (resp. 6) AP deletion the effect was positive (resp. negative) ..." The term 'resp.' is used throughout, I do not know what that means.

Reviewer #2:

Sorry, I use symbols and italics and bold and don't have time for fiddling here. Will email you PDF. YOU NEED TO MAKE IT POSSIBLE TO UPLOAD A PDF.

Yvert Review MSB 2017

This paper compares the fitnesses of over 3500 budding yeast mutants in three conditions, normal medium, high salt, and fluctuations between these two environments, and focuses on the strains whose fitness in the fluctuating environment differs from the average of the value over the two stable environments. I see this as more a paper of record, which points out that examining fitness in

dynamically variable environments complements more traditional fitness measurements and goes on to make a number of useful points, some of which need further examination, rather than being a paper with a single striking conclusion. If points 2-4 were successfully addressed, this work would be of interest to a reasonably wide breadth of the readership of MSB. If not, it seems better suited to PLoS Genetics.

Major points

1) The authors make the useful general point that many deletions show something different from the average response to low and high salt medium when they are exposed to fluctuating environments, but two questions follow: is this surprising and what can we learn from the details. It's easy for reviewers to retrospectively argue that conclusions are unsurprising, but it makes sense to me that there are many genes that have an influence as cells switch between growth conditions and growth rates. The authors' finding that the variance in their pool's fitness increases as the period of the oscillation gets shorter fits this expectation. In detail, my impression is that the results are idiosyncratic, such as the different response to removing different members of the same signaling complex, a conclusion that strikes me as neither surprising nor especially informative.

2) To me, the most surprising result is that there are many genes whose average fitness is > 1.05 in over the two stable environments and most of these have a higher than expected fitness in a fluctuating environment. Given their high average fitness, it seems likely their fitness is >1 in both environments and it would be good to comment on this. I'm already mystified by the finding that there are so many genes whose deletion increases fitness in exponential growth, and if the interpretation is that you can go faster by removing unneeded regulatory machinery, it seems natural to predict that these mutants would show fitness defects in stressful or fluctuating conditions. I think one of the two following additions would substantially improve the paper: 1) Systematically examine the nature of these genes and their fitness in each stable environment and ask whether the behavior of their deletions can be explained, in terms of their fitness advantage in both stable and fluctuating environments, or 2) take a small number and label them with a fluorescent protein and compete them with wild type in a fluctuating environment, taking samples with enough temporal resolution to determine when, during the cycle of environmental oscillation, the mutants have an advantage. As an example of the latter, removing genes like PDE2 and RAS2 is likely to reduce the drop in cAMP levels when cells change environments, suggesting that these mutants would primarily out proliferate wild type in the first cell division after each environmental transition.

3) There is a good deal of evidence that strains from the deletion collection have a variety of mutations beyond those the single gene deletion they are designed to have. If this was my paper, I would want to take four strains that show a fitness advantage in normal conditions, a fitness advantage in high salt, and a larger than expected advantage in the fluctuating conditions, remake each deletion twice in a MATa and twice in a MAT α strain, make all four possible diploids between the pairs of strains and repeat the fitness measurements.

4) The paper is unnecessarily long, and both the introduction and discussion could probably be halved in length while still making the essential points. Shortening both, and reducing the extent of speculations about cancer, will increase the chance that the intended audience will read the paper to the end.

Minor points

Confusing terminology. The parenthetical (resp. ...) appears in many places and its meaning is unclear, e.g. "gene deletion decreased (resp. increased) proliferation in S (resp. N) for all positive regulators of the pathway".

The strict definition of antagonistic pleiotropy is $w_A > 1$ and $w_B < 1$, where A and B are different environments, whereas Fig 3A reports $w_A - w_B$. Do the strains which generate the pie chart in Fig 3B satisfy the strict definition?

The multiple lines in 2B are hard to disentangle from each other. It would help to have the N line in a color that had no similarity to the dashes in the oscillating lines.

If transgressive behavior is the result of a missing gene, two things should be true: adding back the wild type copy should eliminate the transgressive behavior and it should restore fitness to 1. This second point is true for PDE1 but not for TOM7. There are two explanations for this result: TOM7 is haploinsufficient or there are unexpected properties of the strain that are not due to the mutations in TOM7. Testing the effect of adding back a second copy will distinguish these possibilities.

There are a few minor errors in English which could be corrected by having a native speaker read the paper carefully.

Reviewer #3:

The authors measure the fitness of ~3,500 single gene deletion strains grown exponentially in benign (N) or high salt (S) conditions or in fluctuations between the two environments at various frequencies. They find that a large number of gene deletions are inhomogeneous in environmental fluctuations -- which they describe as a significant deviation from the geometric mean of fitnesses in N and S. Inhomogeneity is more pronounced with quicker fluctuations. They next looked specifically at genes with antagonistic pleiotropy (AP) between N and S, and found that most had either consistently positive or consistently negative inhomogeneity across fluctuating regimes, with others having more complex patterns. They next looked at "transgressive" deletions, those with a greater or less fitness during fluctuations than fitness in either N or S, and found a number of them with various cellular processes.

The subject tackled by this manuscript -- how environmental fluctuations impact mutational fitness effects -- is interesting and timely. Deletion screens, and indeed experimental evolution regimes, are typically performed in simple unchanging environments. As the authors point out, this limitation could miss much of the biology concerned with how cells cope with changing environments, thereby leaving a partial view of many cellular processes and gene functions. The authors do find a number of gene deletions that have inhomogeneous fitness effect across fluctuations, but, largely, have little else to say. I am not at all surprised that deletions would be inhomogeneous in their experimental setup (although others may be). In N and S, cells never experience an environmental shift requiring sensing, a cellular response, and ultimately a new steady cell state. Thus any deletions that impact this sense and response would only be detectable in fluctuations (inhomogeneity), with more inhomogeneity at quicker fluctuations. As is, the manuscript reads as a (good) half-finished work, that should go on to explore in more depth the causes of inhomogeneity and transgressivity. I want to let the authors know that I typically hate reviews that ask for a bunch of new experiments and try to avoid it in my reviews, but, in this case I do believe it would be best. One question I would be excited for the authors to explore is what fraction of inhomogeneous gene products are "universal" -- that is, they are inhomogeneous for a large fraction of environmental shifts and represent a core sense and response pathway. The flip side of this is identifying gene deletions that are inhomogeneous to very specific environmental fluctuations. This would, of course, require the authors to perform bar-seq in a number of different environmental fluctuations (although perhaps at only one time interval). However, the outcome would be a much richer data set where the mechanisms of inhomogeneity and of specific gene functions could be more thoroughly explored. The authors should also consider coming up with a better null model (either experimentally or in the analysis) that includes a fitness effect during environmental shifts.

We thank the reviewers for their time and comments which helped improve the manuscript.

Reviewer #1:

In this work, Salignon use the yeast gene-deletion library to compare gene fitness contributions to continuous growth with and without high osmolarity versus growth through cycles of high and low osmolarity. The main contribution is that the authors identify some gene deletion strains whose fitness is different in response to cycling environments compared to the time-averaged sum of fitness in the two steady-state conditions. Some of the differential fitness effects depend on the frequency of osmotic shifts, for reasons that are not really clear.

I find the topic and the motivation interesting, but in the end I'm not sure if the results provide enough of a mechanistic advance for publication in Mol Sys Bio. Most of the results are observational and the authors highlight a few functional categories, but the underlying mechanisms are not clear.

Major points:

1. I found the manuscript difficult to read in places, including where some key methods were discussed. It was unclear which samples were submitted for sequencing - the authors describe cycling cells every 3 hours and at several points in the manuscript they discuss the 6-hour sample - but were barcodes sequenced after only 6h growth? If so, in 96-well plates, this can hardly be more than two doublings and it's hard to imagine that barcodes could be distinguished (since bar-seq experiments rely on generation times to give a real signal).

Text indicates the timing of periods, which is twice larger than the timing of medium changes. Barcodes were sequenced every day, not after only 6h. This is now more explicitly mentioned in results: "populations were sampled at times 0, 24h, 48h and 72h for sequencing" (lines 124-125).

2. Since the assay is so dependent on generation times, it should be more clearly described how many generations cells passed through for each experiment - it is also fundamentally important that cells in different conditions progress through the same number of culture generations to enable comparisons, but it seems here that everything was done based on time, not generations.

Based on optical density measurements, the 3 days duration of the experiment corresponded to 24 generations for wild-type cells in medium N. As in many other studies (e.g. PMIDs 26596348, 24789747, 23103169), we do not know precisely the number of generations for mutant cells but we quantify the deviation from wild-type, in an experiment of fixed duration. Our conclusions are based on two types of computations which gave very consistent results: a statistical inference based on a Generalized Linear Model (GLM), and computation of fitness values (w).

A major advantage of the GLM is that any difference in generation time between conditions is captured by coefficients related to the conditions ($\beta_{i,1}$ and $\beta_{i,2}$ in formula of methods). This way, these differences should not affect the coefficient of interest ($\beta_{i,3}$) which is related to environmental dynamics.

Regarding w values, it is true that fixing $g = 24$ for all conditions in our computation is an approximation, especially for the S medium. Optical density measurements showed that the doubling time of wild-type cells, as well as of the entire pooled population of mutants, is 10% longer in S than in N. So, $g = 21.6$ would be more appropriate when computing w_S . However, we observed that fitness inhomogeneity values did not change significantly when they were recomputed after applying this correction (Supplementary Fig. 5). This is now clearly stated in methods.

3. Data in Figure 2 suggest that cells were cycled at different intervals, but I couldn't find a clear description of that in the Methods. Again it was not clear if the culture went through the same number of generations in each of these experiments, and if not it's hard to know if these are really comparable datasets.

Yes, the experiment included oscillating environments with different periods. This is now better explained in methods, lines 497-500 (please see point 5 below). As for number of generations, please see point 2 above.

Minor points:

4. On page 9, the authors first describe that nearly a third of the yeast gene deletion strains deviate from the time-averaged expectation, but this is at an FDR of 0.2. Most of the manuscript focuses on a more stringent list at a reasonable FDR, but then in the Discussion they refer back to this very large dataset at a relaxed FDR. Best to focus on the more stringent gene set, which is plenty of genes for analysis; there seems to be no good reason to relax the FDR cutoff.

Yes, the reason to relax the FDR is to estimate the overall amount of significant genes in the genome, and this is reported in the results section. The revised discussion now mentions only the stringent list.

5. Page 10: under 'Impact of environmental dynamics on mutants proliferation': this is the first the authors mention different oscillation time scales, and the text implies they are doing something computational - but data in Fig 2 looks like they did different cycling times. This should be more clearly presented in the text and the Methods.

We did not do it computationally but experimentally. The different oscillating periods are mentioned at the very beginning of the results section (page 7). The revised text in page 10 now clarifies that "Our experiment included conditions with 4 periods larger than 6h. For each period, we computed...". The revised methods read "The salt experiment included 7 culture conditions (N, S, NS6, NS12, NS18, NS24, NS42), where N and S were steady environments, and NSx was an environment alternating between N and S with a period of x hours. The methionine experiment also included 7 culture conditions (steady M0, steady M1, and alternating conditions with periods of 6, 12, 18, 24 or 42 hours)."

6. Some of the terminology is confusing, e.g. this sentence on P12: "... For 33 (resp. 6) AP deletion the effect was positive (resp. negative) ...". The term 'resp.' is used throughout, I do not know what that means.

Sorry for this confusing writing. The sentence now reads "The effect was positive at all periods for 33 AP deletions, and negative at all periods for 6 AP deletions."

Reviewer #2:

Sorry, I use symbols and italics and bold and don't have time for fiddling here. Will email you PDF. YOU NEED TO MAKE IT POSSIBLE TO UPLOAD A PDF.

Yvert Review MSB 2017

This paper compares the fitnesses of over 3500 budding yeast mutants in three conditions, normal medium, high salt, and fluctuations between these two environments, and focuses on the strains whose fitness in the fluctuating environment differs from the average of the value over the two stable environments. I see this as more a paper of record, which points out that examining fitness in dynamically variable environments complements more traditional fitness measurements and goes on to make a number of useful points, some of which need further examination, rather than being a paper with a single striking conclusion. If points 2-4 were successfully addressed, this work would be of interest to a reasonably wide breadth of the readership of MSB. If not, it seems better suited to PLoS Genetics.

Major points

1) *The authors make the useful general point that many deletions show something different from the average response to low and high salt medium when they are exposed to fluctuating environments, but two questions follow: is this surprising and what can we learn from the details. It's easy for reviewers to retrospectively argue that conclusions are unsurprising, but it makes sense to me that there are many genes that have an influence as cells switch between growth conditions and growth rates.*

We see our study as a genomic screen. As for any genetic screen, before it is done scientists guess that many genes would have an influence; after it is achieved, a list of genes is obtained that constitutes knowledge and a scientific basis (entry points, or keys) for further studies. Note that, as now reported in the revised manuscript, the expectation of many genes showing inhomogeneity is correct for both salt stress and methionine oscillations, but the extent and direction of inhomogeneity are different (Fig. 6), which would be hard to predict before seeing the data.

The authors' finding that the variance in their pool's fitness increases as the period of the oscillation gets shorter fits this expectation.

Yes. And, consistently with the lower extent of inhomogeneity, we do not see this variance increase for methionine oscillations (Fig. 6B).

In detail, my impression is that the results are idiosyncratic, such as the different response to removing different members of the same signaling complex, a conclusion that strikes me as neither surprising nor especially informative.

We agree with the reviewer that finding differences between elements of the same pathway is not necessarily surprising, but we are also happy to identify and quantify these differences. For example, it is hard to imagine how one could predict, before our study, that the fitness of *rim21Δ* but not of *rim20Δ* would be inhomogeneous in salinity oscillations. We provide this information, at the genomic scale.

2) To me, the most surprising result is that there are many genes whose average fitness is > 1.05 in over the two stable environments and most of these have a higher than expected fitness in a fluctuating environment. Given their high average fitness, it seems likely their fitness is >1 in both environments and it would be good to comment on this. I'm already mystified by the finding that there are so many genes whose deletion increases fitness in exponential growth, and if the interpretation is that you can go faster by removing unneeded regulatory machinery, it seems natural to predict that these mutants would show fitness defects in stressful or fluctuating conditions. I think one of the two following additions would substantially improve the paper: 1) Systematically examine the nature of these genes and their fitness in each stable environment and ask whether the behavior of their deletions can be explained, in terms of their fitness advantage in both stable and fluctuating environments, or 2) take a small number and label them with a fluorescent protein and compete them with wild type in a fluctuating environment, taking samples with enough temporal resolution to determine when, during the cycle of environmental oscillation, the mutants have an advantage. As an example of the latter, removing genes like PDE2 and RAS2 is likely to reduce the drop in cAMP levels when cells change environments, suggesting that these mutants would primarily out proliferate wild type in the first cell division after each environmental transition.

Regarding suggestion 1), we have explicitly compared inhomogeneity with the fitness in steady conditions (Fig. 3 of the revised manuscript). As anticipated by the reviewer, most genes with average fitness > 1.05 indeed have a higher-than-expected fitness in a fluctuating environment (most red dots are above 1). Very interestingly, this also revealed a set of genes that we had missed in our initial analysis, and we thank the reviewer for his/her suggestion. These genes have high inhomogeneity and an average fitness ~ 1 (blue dots). Deletions of these confer >1 fitness in N but ≤1 fitness in S. Annotations in this set were enriched for "osmosensing and response", because of the 6 genes labeled on the figure (Supplementary Table 2C). We interpret these genes as being early triggers of the cellular response in wild-type cells, which slow their division as compared to mutant cells where these triggers are defective. Notably, not all genes known as osmosensors/responders behaved this way (e.g. RCK2, HOT1), which further illustrates how guessing which deletions are homogeneous or not is difficult without experimental data. These novel observations are reported in the revised text (lines 219-228).

In addition, under methionine oscillations, high-expected fitness is not associated with "even-higher-than-expected". See *rrt12Δ/Δ* mutant on Fig. 6A for example. Thus, the overall effect seen under salt oscillations is not a general rule.

Regarding suggestion 2), environmental transitions (3h) are on the same time-scale as the doubling time (~3h). It is therefore challenging, even with FACS-based cell counting, to detect out-proliferation at higher temporal resolution. We are now setting up collaborations to track the cell response in microfluidics, in particular mutants carrying appropriate reporters. Obtaining reliable answers will take time. As for RAS2, although BAR-seq picked it as transgressive in the pooled population, individual assay by FACS revealed a rather homogeneous fitness in the fluctuating regime (Supplementary Fig. 2).

3) *There is a good deal of evidence that strains from the deletion collection have a variety of mutations beyond those the single gene deletion they are designed to have. If this was my paper, I would want to take four strains that show a fitness advantage in normal conditions, a fitness advantage in high salt, and a larger than expected advantage in the fluctuating conditions, remake each deletion twice in a MATa and twice in a MATa strain, make all four possible diploids between the pairs of strains and repeat the fitness measurements.*

We agree that additional mutations of the deletion strains can be an issue, which is why we applied complementation assays on two of them. Complementation has the advantage to rule out regulatory perturbations of neighboring genes in *cis*. A likely example of this is the *rpl40AA* deletion which is associated with transgressive fitness (Supplementary Fig. 2) and which not only removes a coding sequence but also targets the 3'UTR of the upstream gene *SLN1*, an important osmosensor regulating the HOG pathway.

Complementation also has a limit: if the phenotype is not reversed, integrity and proper expression of the inserted wild-type copy must be verified. The experiment suggested by the reviewer is a thorough additional control in cases where complementation does not validate the implication of the gene product: it would not discriminate *cis*-acting effects but would indeed rule out the involvement of undesired additional mutations. We remain open to run such *de novo* deletion controls if the editor considers they are essential.

4) *The paper is unnecessarily long, and both the introduction and discussion could probably be halved in length while still making the essential points. Shortening both, and reducing the extent of speculations about cancer, will increase the chance that the intended audience will read the paper to the end.*

We have shortened the introduction by ~20% and the discussion by ~30% of their initial length. We also shortened the discussion paragraph covering possible implications in cancer.

Minor points

Confusing terminology. The parenthetical (resp. ...) appears in many places and its meaning is unclear, e.g. "gene deletion decreased (resp. increased) proliferation in S (resp. N) for all positive regulators of the pathway".

Thank you. This sentence (lines 142-144), as well as sentences in lines 240-241 and 640-641 have been corrected.

The strict definition of antagonistic pleiotropy is $w_A > 1$ and $w_B < 1$, where A and B are different environments, whereas Fig 3A reports $w_A - w_B$. Do the strains which generate the pie chart in Fig 3B satisfy the strict definition?

Yes, they do. This is now better explained in text (lines 230-250) and their total number is mentioned in legend (now Fig. 4A).

The multiple lines in 2B are hard to disentangle from each other. It would help to have the N line in a color that had no similarity to the dashes in the oscillating lines.

The color has been changed.

If transgressive behavior is the result of a missing gene, two things should be true: adding back the wild type copy should eliminate the transgressive behavior and it should restore fitness to 1. This second point is true for PDE1 but not for TOM7. There are two explanations for this result: TOM7 is haploinsufficient or there are unexpected properties of

the strain that are not due to the mutations in TOM7. Testing the effect of adding back a second copy will distinguish these possibilities.

Note that the magnitude of the behavior is much weaker for TOM7 than for PDE2 and fitness deviation from 1 is therefore more visible (different y-axis scale between Fig. 5F and 5G). We believe that elimination of transgressivity reliably assigns the effect to the gene product. The remaining deviation from 1, even in steady conditions, could indeed result from the possibilities mentioned by the reviewer. It could also result from the genomic (regulatory) context of the ectopic integration site of the wild-type copy of the gene. A two-copies complementation assay may therefore not be enough to fully understand all fitness values but the focus of the experiment was on transgressivity itself.

There are a few minor errors in English which could be corrected by having a native speaker read the paper carefully.

We are grateful to native speaker Steve Garvis for English corrections of the revised manuscript.

Reviewer #3:

The authors measure the fitness of ~3,500 single gene deletion strains grown exponentially in benign (N) or high salt (S) conditions or in fluctuations between the two environments at various frequencies. They find that a large number of gene deletions are inhomogenous in environmental fluctuations -- which they describe as a significant deviation from the geometric mean of fitnesses in N and S. Inhomogeneity is more pronounced with quicker fluctuations. They next looked specifically at genes with antagonistic pleiotropy (AP) between N and S, and found that most had either consistently positive or consistently negative inhomogeneity across fluctuating regimes, with others having more complex patterns. They next looked at "transgressive" deletions, those with a greater or less fitness during fluctuations than fitness in either N or S, and found a number of them with various cellular processes.

The subject tackled by this manuscript -- how environmental fluctuations impact mutational fitness effects -- is interesting and timely. Deletion screens, and indeed experimental evolution regimes, are typically performed in simple unchanging environments. As the authors point out, this limitation could miss much of the biology concerned with how cells cope with changing environments, thereby leaving a partial view of many cellular processes and gene functions. The authors do find a number of gene deletions that have inhomogenous fitness effect across fluctuations, but, largely, have little else to say. I am not at all surprised that deletions would be inhomogenous in their experimental setup (although others may be). In N and S, cells never experience an environmental shift requiring sensing, a cellular response, and ultimately a new steady cell state. Thus any deletions that impact this sense and response would only be detectable in fluctuations (inhomogeneity), with more inhomogeneity at quicker fluctuations. As is, the manuscript reads as a (good) half-finished work, that should go on to explore in more depth the causes of inhomogeneity and transgressivity. I want to let the authors know that I typically hate reviews that ask for a bunch of new experiments and try to avoid it in my reviews, but, in this case I do believe it would be best.

Figure 3 of the revised manuscript now makes it more apparent that some deletions impacting the response show inhomogeneity (CIN5, SSK1...), while others do not (RCK2, HOT1...). We share the reviewer's motivation to dig into mechanistic processes explaining the behaviour of some of the mutants. In the case of periodic environments, obtaining mechanistic insights will need to follow the response dynamics in microfluidics time-course data, which corresponds to significant amounts of time, efforts and money. We are currently teaming up with a physics laboratory having the necessary skills and expertise, and we are also raising funds to run this characterization. For now, we provide the results of our screen so that the entire community can follow up on specific mutants of interest.

One question I would be excited for the authors to explore is what fraction of inhomogenous gene products are "universal" -- that is, they are inhomogenous for a large fraction of environmental shifts and represent a core sense and response pathway. The flip side of this is identifying gene deletions that are inhomogenous to very specific environmental fluctuations. This would, of course, require the authors to perform bar-seq in a number of different

environmental fluctuations (although perhaps at only one time interval). However, the outcome would be a much richer data set where the mechanisms of inhomogeneity and of specific gene functions could be more thoroughly explored.

To examine the generality/specificity of fitness inhomogeneity, we have performed another genomic profiling in periodic fluctuations of extracellular methionine concentrations (between 0 and 1 mM), a condition that is unrelated to salt stress. This experiment showed that:

- i) In contrast to salt oscillations, methionine oscillations do not cause higher-than-expected fitness of numerous mutants. Rather, methionine oscillations tend to reduce the fitness of many mutants below expectation (Fig. 6A)
- ii) Accordingly, the increase of genetic variance that we saw at short periods of salt stress is not visible at short periods of methionine fluctuations (Fig. 6B)
- iii) There is no overall correlation between inhomogeneities seen in periodic salt stress and in methionine concentration (Fig. 3C).
- iv) A handful of genes displayed higher-than-expected fitness in both experiments (e.g. RCM1, Fig. 6D). Annotation of these genes did not point to a common pathway or mechanism.
- v) One mutant showed a strong reverse inhomogeneity (RRT12, Fig. 6C)

The fact that salt stress causes a widespread genomic twist whereas methionine oscillations have milder effects is likely due to different target sizes in the genome. Salt stress is known to induce global changes in cells (cell shrinkage, osmotic changes, ionic disequilibrium) which differs from the specificity of a metabolic pathway (although methionine, as a sulfur amino-acid, is connected to AdoMet and therefore to methylations in general). This conclusion has been added to the discussion.

The authors should also consider coming up with a better null model (either experimentally or in the analysis) that includes a fitness effect during environmental shifts.

We are now planning to run asymmetric fluctuations (e.g. short epoch of S, so that $N > S$ are poorly selective and $S > N$ are highly selective). This, together with microfluidics experiments that we are now designing, should help determine which transitions are important for fitness inhomogeneity, and what happens and when at the molecular level after the relevant transitions.

Thank you again for submitting your work to Molecular Systems Biology. We have now heard back from the referee who accepted to evaluate the revised study. As you will see, the referee raises some final comments that we would kindly ask you to address with suitable amendments in the text.

We would be grateful if you could also address the following points:

Main manuscript files

- Please add up to five keywords.
- Please update the reference format from numerical to alphabetical to match the MSB reference style. <http://msb.embopress.org/authorguide#referencesformat>
- Please replace the manuscript PDF with a word-doc file.

Appendix

- Please rename 'Supplementary Tables' -> 'Appendix'.
- Please add a Table of Contents with page numbers.
- Please integrate the individual supplementary figures in the Appendix file.
- Please move the supplementary figure legends and table legends from the manuscript file to the Appendix.
- Please rename the tables, figures and legends and update the callouts to Appendix table S[n] and Appendix Fig. S[n].

Datasets

- Please rename them from Supplementary Datasets to Dataset EV[n] and update the callouts to both datasets accordingly.
- The dataset file were uploaded twice. Please double check if there was a versioning issue or some mistakes and upload only the version that should be published.

Callouts to main figures

- Please add callouts to all panels in the main figures (Callouts to fig. 3A-D and fig. 4C seem to be missing).

REVIEWER REPORT

Reviewer #3:

Major points

-- The authors did address my main criticism, although minimally, by looking at an additional fluctuation (methionine availability), finding that few inhomogeneous KOs overlap between the two sets. A richer data set would have been to look at a few fluctuations, particularly others that induce a stress response, where some overlaps might be expected and dissected. However, the new experiment does offer some insight into the biology, and I believe it is a sufficient improvement for publication in MSB.

-- The fitness measure borrowed from Qian et al. is not a robust fitness measure because it does account for a changing mean fitness. That is, as the fittest lineages in the population expand, the mean fitness of the population will increase, causing lineages to "bend down" as competition gets tougher. This effect is apparent in several places, most notably Fig 2b. Because of this effect, endpoint enrichment is an unstable fitness measure that will change depending on the environment and fitness distribution of the population. For a reasonable alternative, see supp materials of Nat Commun. 2017 May 25;8:15586. I will note that I don't believe that a reanalysis is likely to change the major conclusions of this work, but it is the best practice.

-- The section on fitness variation between mutants is not well explained in either the results (lines 207- 215) or the methods, where some variables are undefined. I had to read this several times to understand what the authors are doing and why.

-- Line 425 - "Negative regulators of the pathway, including PDE2, are recurrent targets for de novo mutations in yeast populations evolving in steady experimental conditions." This is in incorrect characterization, and what the authors get wrong is important. In the referenced experiment and other batch evolutions, cells are fluctuating between nutrient rich and nutrient depleted environments. Results in this manuscript suggest that mutations in cAMP/PKA may produce insensitivity to nutrient depletion stress allowing for extra growth near saturation. An analogy to S in this case may be a prolonged starvation, where these mutants could perform much worse than WT.

-- I agree with another reviewer that the cancer discussion is tenuous and the manuscript would be stronger without it

Minor points

-- Line 92 - - "time-average of its fitness in each of the alternating condition" -- delete "of the"

-- Line 251 - "exacerbate" is not the right word, since this has a negative connotation. It is used again on line 342.

-- Line 260 "This reveals that environmental oscillations on short time scales can twist natural selection in favour of a subset of mutations on the long term." This is speculation without proof. Lines 260 - 268 are better left to the discussion.

-- The paragraph that starts on line 302 is poorly written as compared to the rest of the document. I would suggest having it revised by a native English speaker.

-- Many typographical errors in the parenthetical on line 344

-- line 360 "our survey provides a genome-scale view of natural selection in periodic environments". No it doesn't. There is no natural selection in these experiments.

-- line 440 -- change "probably results from" to "could be due to"

We thank the reviewer for his/her helpful additional comments.

Reviewer #3:

Major points

-- The authors did address my main criticism, although minimally, by looking at an additional fluctuation (methionine availability), finding that few inhomogeneous KOs overlap between the two sets. A richer data set would have been to look at a few fluctuations, particularly others that induce a stress response, where some overlaps might be expected and dissected. However, the new experiment does offer some insight into the biology, and I believe it is a sufficient improvement for publication in MSB.

Thank you for this feedback.

-- The fitness measure borrowed from Qian et al. is not a robust fitness measure because it does account for a changing mean fitness. That is, as the fittest lineages in the population expand, the mean fitness of the population will increase, causing lineages to "bend down" as competition gets tougher. This effect is apparent in several places, most notably Fig 2b. Because of this effect, endpoint enrichment is an unstable fitness measure that will change depending on the environment and fitness distribution of the population. For a reasonable alternative, see *supp materials of Nat Commun. 2017 May 25;8:15586*. I will note that I don't believe that a reanalysis is likely to change the major conclusions of this work, but it is the best practice.

The "bending down" seen in Fig. 2B could be due to a change of population mean fitness, but we sometimes see it in situations where only 2 strains (WT and mutant) are assessed by flow-cytometry (e.g. Fig 5B & 5D). We therefore believe that something else is also going on, at the level of the biology of adaptation. In addition, the w estimate from Qian et al. gives a relative between one mutant and a 'meta-wild-type' estimated from several pseudogenes. The consequence on w of an increase in pop mean fitness is not simple to anticipate, because both the mutant and the meta-wild-type will be affected. The alternative measure of Nat Commun 2017 is very interesting and we now cite it in the methods.

-- The section on fitness variation between mutants is not well explained in either the results (lines 207- 215) or the methods, where some variables are undefined. I had to read this several times to understand what the authors are doing and why.

We now explain in line 210 that " Distinguishing the genetic variance from the non-genetic variance was possible because of the presence of replicates in our experimental design"; methods now specify that V_E is the residual variance that reflects inter-replicates variability, and that calling it the 'environmental variance' would be confusing in the context of this study.

-- Line 425 - "Negative regulators of the pathway, including PDE2, are recurrent targets for de novo mutations in yeast populations evolving in steady experimental conditions." This is in incorrect characterization, and what the authors get wrong is important. In the referenced experiment and other batch evolutions, cells are fluctuating between nutrient rich and nutrient depleted environments. Results in this manuscript suggest that mutations in cAMP/PKA may produce insensitivity to nutrient depletion stress allowing for extra growth near saturation. An analogy to S in this case may be a prolonged starvation, where these mutants could perform much worse than WT.

Yes, this is an important similarity between these studies and what we did. We corrected the sentence by "evolving in serial transfer experiments".

-- I agree with another reviewer that the cancer discussion is tenuous and the manuscript would be stronger without it

There is currently massive research on the spectrum of mutations found in cancers and yet, many cancer biologists are still not familiar with the evolutionary properties that shape the spectrum of mutations that are observed. In particular, the importance of environmental conditions is often underestimated (probably because it is very hard to assess). We'd like to keep this section as an

opening to this community. To make it less speculative, we have removed the possible impact on PDE mutations.

Minor points

-- Line 92 -- *"time-average of its fitness in each of the alternating condition" -- delete "of the"*

Has been modified.

-- Line 251 - *"exacerbate" is not the right word, since this has a negative connotation. It is used again on line 342.*

We changed it by "heighten"

-- Line 260 *"This reveals that environmental oscillations on short time scales can twist natural selection in favour of a subset of mutations on the long term." This is speculation without proof. Lines 260 - 268 are better left to the discussion.*

We moved this speculation to the discussion (lines 375-377).

-- *The paragraph that starts on line 302 is poorly written as compared to the rest of the document. I would suggest having it revised by a native English speaker.*

English of this paragraph was corrected by a native speaker (Steve Garvis).

-- *Many typographical errors in the parenthetical on line 344*

Fixed.

-- *line 360 "our survey provides a genome-scale view of natural selection in periodic environments". No it doesn't. There is no natural selection in these experiments.*

We changed it by "of the selection of mutations".

-- *line 440 -- change "probably results from" to "could be due to"*

Done.

Corresponding Author Name: Gaël YVERT

Manuscript Number: MSB-17-7823R